# Megakaryocytes assemble a three-dimensional cage of extracellular matrix that controls their maturation and anchoring to the vascular niche

Claire Masson[1†], Cyril Scandola[2†], Jean-Yves Rinckel[1], Fabienne Proamer[1], Emily Janus-Bell[1], Fareeha Batool[1], Naël Osmani[3], Jacky G Goetz[3], Léa Mallo[1], Nathalie Brouard[1], Catherine Leon[1], Alicia Bornert[1], Renaud Poincloux[4], Olivier Destaing[5], Alma Mansson[6], Hong Qian[6], Maxime Lehmann[1], Anita Eckly[1*]

[1]Université de Strasbourg, INSERM, Strasbourg, France; [2]Institut Pasteur, Université Paris Cité, Paris, France; [3]INSERM UMR_S1109, Tumor Biomechanics Lab, Université de Strasbourg, Fédération de Médecine Translationnelle de Strasbourg (FMTS),, Strasbourg, France; [4]Institut de Pharmacologie et de Biologie Structurale, Université de Toulouse, Toulouse, France; [5]Institute for Advanced Biosciences, Centre de Recherche Université Grenoble Alpes, La Tronche, France; [6]Center for Hematology and Regenerative Medicine (HERM), Department of Medicine Hiddinge, Karolinska University Hospital, Karonlinska Institute, Stockholm, Sweden

**\*For correspondence:**
Anita.Michel@efs.sante.fr

[†]These authors contributed equally to this work

**Competing interest:** The authors declare that no competing interests exist.

## eLife Assessment

In this revised version, the authors provide a thorough investigation of the interaction of megakaryocytes (MK) with their associated extracellular matrix (ECM) during maturation; they provide **compelling** evidence that the existence of a dense cage-like pericellular structure containing laminin γ1 and α4 and collagen IV is key to fixing the perisinusoidal localization of MK and preventing their premature intravasation. Adhesion of MK to this ECM cage is dependent on integrin beta1 and beta3 expressed by MK. This strong conclusion is based on the use of state-of-the art techniques such f primary murine bone marrow MK cultures, mice lacking ECM receptors, namely integrin beta1 and beta3 null mice, as well as high-resolution 2D and 3D imaging. The study provides **valuable** insight into the role of cell-matrix interactions in MK maturation and provides an interesting model with practical implications for the fields of hemostasis and thrombosis.

**Abstract** Megakaryocytes, the progenitor cells of blood platelets, play a crucial role in hemostasis by residing in the bone marrow and ensuring continuous platelet production. Unlike other hematopoietic cells, megakaryocytes do not enter the blood circulation intact. They remain anchored within the bone marrow while extending cytoplasmic protrusions called proplatelets through the sinusoidal endothelial barrier. These proplatelets subsequently fragment into functional platelets. This unique process of intravasation facilitates efficient platelet production while maintaining the megakaryocyte cell body within the bone marrow niche, thus preventing potential thrombotic complications. How the extracellular matrix (ECM) influences the delicate balance between megakaryocyte retention and proplatelet extension remains largely unknown. Here, we investigate the spatial organization and functional role of ECM components in the megakaryocyte vascular niche of mice bone marrow. Our findings reveal that laminin and collagen IV form three-dimensional (3D) ECM cages encompassing megakaryocytes and anchor them to the sinusoidal basement

membrane. Gene deletion shows the existence of laminin α4 in the ECM cage that is necessary to maintain megakaryocyte-sinusoid interactions. Notably, megakaryocytes actively contribute to the ECM cage assembly; β1/β3 integrin knockout weakens these structures, increasing intravasation and entire megakaryocyte entry into circulation. The retention of megakaryocytes by these 3D ECM cages depends on dynamic remodeling processes. Inhibition of ECM proteolysis results in denser cage formation, increasing the frequency of immature megakaryocytes with impaired demarcation membrane system (DMS) development. Thus, the ECM cage represents a novel concept of an active and dynamic 3D microenvironment that is continuously remodeled and essential for maintaining megakaryocyte perivascular positioning. This specific microarchitecture guides megakaryocyte maturation and intravasation, underscoring the critical role of ECM microarchitecture and dynamics in megakaryocyte function.

## Introduction

Megakaryocytes, the precursors of blood platelets, differentiate from hematopoietic stem cells and mature in a complex bone marrow microenvironment. Megakaryocytes are large, highly polyploid cells that contain a complex invaginated network of membranes, known as the demarcation membrane system (DMS), which serves as a membrane reservoir for platelet production. In contrast to other hematopoietic cells, megakaryocytes exhibit a unique platelet production process. They do not enter the bloodstream intact. Instead, they remain anchored in the bone marrow and extend cytoplasmic protrusions called proplatelets through the sinusoidal endothelial barrier. These protrusions protrude into the lumen of bone marrow sinusoids, where they undergo fragmentation to release individual platelets into the circulation. This distinctive mechanism allows for the efficient production and release of platelets while maintaining the megakaryocyte cell body within the bone marrow microenvironment, thus reducing the risk of thrombotic complications (*Boscher et al., 2020*; *Stone et al., 2022*). The remaining cell body, composed of a nucleus surrounded by a thin rim of cytoplasm, is ultimately phagocytosed in the stroma (*Radley and Haller, 1983*).

Megakaryocytes are strategically located at the interface between the bone marrow and the blood circulation, specifically at the parasinusoidal region (*Lichtman et al., 1978*; *Stegner et al., 2017*). Intravasation, the coordinated passage of megakaryocyte fragments through the sinusoidal barrier, requires an original, complex, and dynamic adaptation of megakaryocytes to highly different microenvironments, both mechanically and biologically. They are positioned next to the endothelial lining and its underlying basement membrane, in equilibrium between the constrained 3D environment of the bone marrow and the fluid environment of the blood. In this context, they are in contact with two types of ECM organization: basement membrane and interstitial ECM.

Megakaryocytes actively influence their structural microenvironment through several ECM remodeling mechanisms, including the endocytosis of plasma proteins like fibrinogen, and the synthesis and release of extracellular matrix (ECM) components such as fibronectin, laminin, and type IV collagen (*Abbonante et al., 2017*; *Handagama et al., 1987*; *Malara et al., 2014*). Additionally, they produce ECM-modifying proteins, including matrix metalloproteinases (MMPs), tissue inhibitors of metalloproteinases (TIMPs), and lysyl oxidase (LOX), which facilitate the continuous renewal and regulation of the matrix (*Eliades et al., 2011*; *Malara et al., 2018*; *Villeneuve et al., 2009*). Furthermore, we have recently discovered that megakaryocytes could remodel the substrate-bound fibronectin matrix into basal fibrillar structures surrounding the cells. The extent of fibrillogenesis depended on the stiffness of the substrate and relied on both β1 and β3 integrins (*Guinard et al., 2023*). This intricate interplay between megakaryocytes and their surrounding ECM remodeling demonstrates the dynamic nature of the vascular niche and highlights its critical role in platelet production.

The ECM surrounding megakaryocytes plays a crucial role in their development and function. Numerous studies have identified essential components of this matrix, including collagen IV, fibronectin, fibrinogen, and laminin (*Larson and Watson, 2006*; *Malara et al., 2014*; *Semeniak et al., 2016*; *Susek et al., 2018*). The presence and function of collagen I fibers remain a subject of ongoing debate, with some researchers proposing that they guide megakaryocyte proplatelets toward sinusoids (*Oprescu et al., 2022*). In vitro studies have shown that while collagen I may inhibit proplatelet formation, other ECM components, like collagen IV, fibrinogen, and fibronectin, can facilitate this process (*Balduini et al., 2008*; *Malara et al., 2014*; *Sabri et al., 2004*). The diverse and sometimes

contradictory roles of these ECM components in megakaryocyte behavior underscore the complexity of this research area and highlight the need for further investigation, particularly in vivo.

The present study provides new insight into the spatial organization of the ECM that envelops megakaryocytes, elucidating its specific architecture and molecular mechanisms governing its dynamics and function. We performed advanced 2D and 3D analyses of mouse bone marrow and identified a novel ECM cage containing laminin chains ϒ1 and α4, and collagen IV that anchored megakaryocytes to the sinusoidal basement membrane. Deleting laminin α4 impairs the connections between megakaryocytes and sinusoids. Megakaryocytes act actively in forming and maintaining this ECM cage, relying on their integrins β1 and β3. Indeed, deleting these integrins causes significant impairment of the cage's integrity, resulting in increased intravasation of megakaryocytes and an unexpected release of intact megakaryocytes into the bloodstream. Moreover, in vivo, matrix metalloproteinases (MMPs) inhibition reveals that dynamic ECM remodeling is crucial for maintaining the cage structure and supporting megakaryocyte development. In conclusion, our study unveils a 3D ECM cage that physically stabilizes megakaryocytes within their vascular niche. This structure facilitates the physiological regulation of megakaryocyte maturation and intravasation at the bone marrow-bloodstream interface.

## Results

### Laminin and collagen IV create a 3D cage around sinusoid-associated megakaryocytes

The organization of the ECM around megakaryocytes in the vascular niche is not well understood due to challenges in in vivo observation. To investigate this, we utilized immunofluorescence (IF) microscopy on ultrathin bone marrow cryosections (250 nm thickness). This approach offers superior axial resolution compared to traditional confocal microscopy, enabling high-resolution localization of ECM components. Megakaryocytes were visualized using antibodies against GPIbβ, while sinusoids were identified with antibodies against FABP4 (*Figure 1A*). Our findings revealed that laminin ϒ1 chains and collagen IV delineated the basement membrane underlying the endothelium of sinusoids and the outer contour of mature megakaryocytes (*Figure 1A*, *Figure 1—figure supplement 1Aa*). Adjacent cells exhibited granular staining for laminin ϒ1 and collagen IV, which may indicate a potential contribution to the ECM constitution (arrow in *Figure 1—figure supplement 1Ag-h*). Fibronectin was detected in the basement membrane and around megakaryocytes, while fibrinogen was more widespread, associated with the basement membrane, the megakaryocyte surface, and intracellular granules (*Figure 1—figure supplement 1Ab-c*). Von Willebrand factor (VWF) signal was restricted to the alpha granules of megakaryocytes (*Figure 1—figure supplement 1Ad*). We observed no detectable signals for type I and III collagen around megakaryocytes or sinusoids despite using antibodies validated on positive controls (*Figure 1—figure supplement 1Ae-f*, *Figure 1—figure supplement 1B–C*).

Having established the expression of laminin ϒ1, collagen IV, fibronectin, and fibrinogen in the direct megakaryocyte microenvironment, their spatial organization was next investigated using 3D imaging of whole-mount bone marrow preparations (*Figure 1B*). Analysis of maximum Z-stack projections revealed that laminin ϒ1 and collagen IV formed a reticular 3D ECM cage completely enveloping megakaryocytes and extending radially from the sinusoidal basement membrane (*Figure 1B–C*). With this approach, the basement membrane-ECM cage connections were clearly visible when examining the full z-stack series. Examples are shown in *Videos 1 and 2* and in *Figure 1—figure supplement 1*. Quantification showed that almost all megakaryocytes near sinusoids (sMK, 92.8 ± 3.3%) have a laminin cage, compared to only 11.4 ± 4.8% of megakaryocytes in the parenchyma (pMK; *Figure 1—figure supplement 1F–G*). Parenchymal megakaryocytes (pMK), which represented only 18 ± 1.3% of all megakaryocyte population (*Figure 1—figure supplement 1H*), were instead surrounded by a sparse thin network of laminin ϒ1 (*Figure 1—figure supplement 1F*). Although we cannot exclude that ECM cage can be formed on its own, our data suggests that ECM cage assembly may require interactions between megakaryocytes and the sinusoidal basement membrane. Parenchymal megakaryocytes (pMK), which represented only 18 ± 1.3% of all megakaryocyte population (*Figure 1—figure supplement 1H*), were instead surrounded by a sparse thin network of laminin ϒ1 (*Figure 1—figure supplement 1F*). The ECM cage was present at all stages of megakaryocyte maturation, including megakaryocytes with proplatelet extension (*Figure 1—figure supplement 1I–J*). Remarkably, after

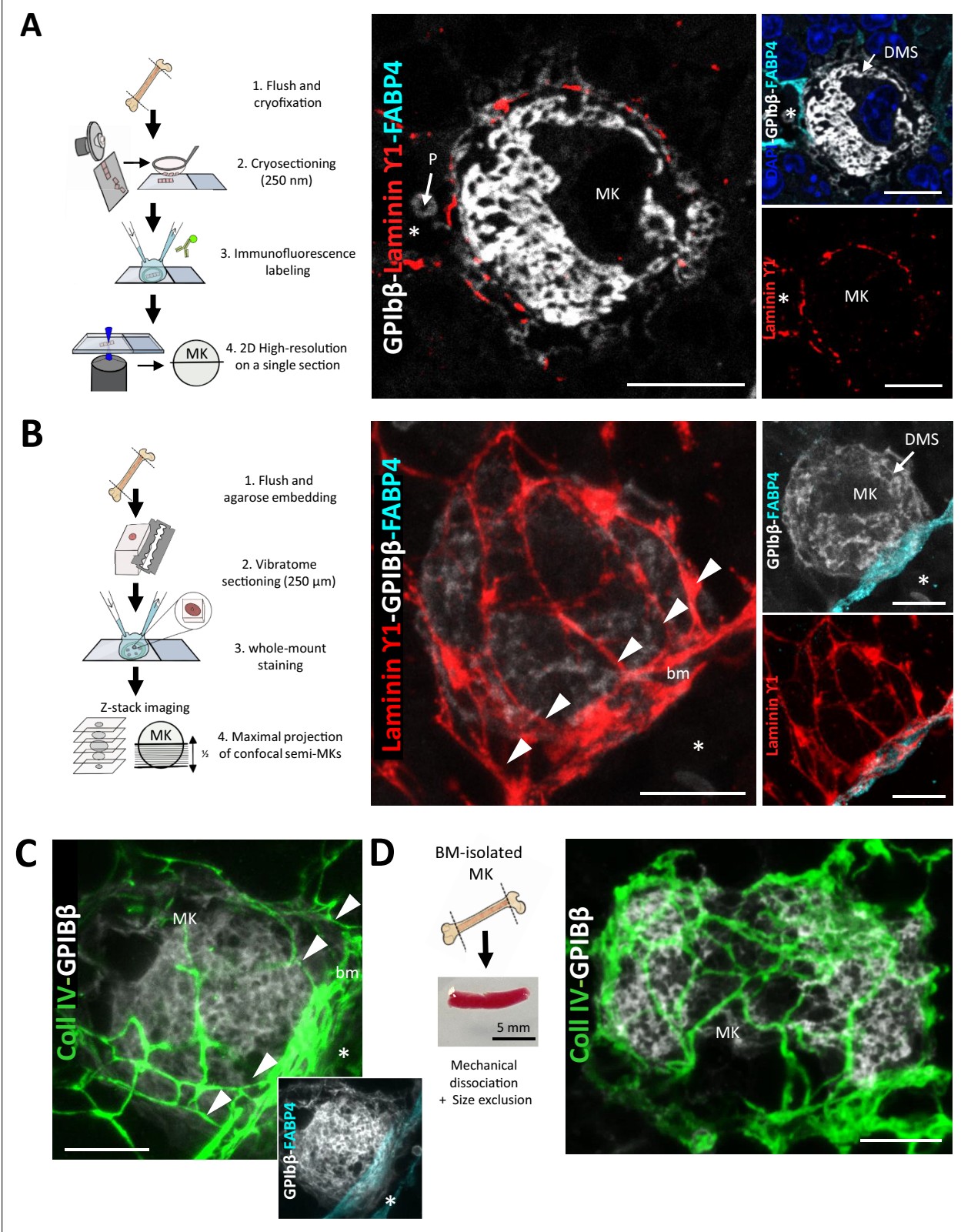

**Figure 1.** Laminin ϒ1 and collagen IV form 3D cages of ECM for megakaryocytes, directly connected to the sinusoidal basement membrane. (**A**) Left panel. Schematic representation of the experimental workflow for 2D imaging of immunostained bone marrow cryosections from WT mice. Confocal imaging is performed on single ultrathin sections with an axial resolution of 250 μm. Right panel. Representative 2D images of a sinusoid-associated megakaryocyte immunostained for laminin ϒ1 (red), GPIbβ (white), and FABP4 (cyan). Cell nuclei are visualized with DAPI (blue; from one out of three

*Figure 1 continued on next page*

*Figure 1 continued*

independent IF experiments). (**B**) <u>Left panel</u>. Schematic representation of the experimental workflow for 3D analysis of whole-mount bone marrow preparations from WT mice. A stack of confocal images covering half the depth of the megakaryocyte is acquired and then z-projected to create a maximum projection image. <u>Right panel</u>. Representative maximal projection images of sinusoid-associated megakaryocyte immunostained for laminin ϒ1 (red), GPIbβ (white), and FABP4 (cyan; from one out of three independent IF experiments). (**C**) Representative maximal projection images of sinusoid-associated megakaryocyte immunostained for collagen IV (green) and GPIbβ (white). The inset image shows the FABP4 (cyan) and megakaryocyte (white) immunostaining. (**D**) Bone marrow-isolated megakaryocyte maintaining an ECM cage. <u>Left panel</u>. Schematic of the experimental procedure used to isolate mouse bone marrow megakaryocytes. <u>Right panel</u>. Maximal projection 3D images showing the persistence of the ECM cage (collagen IV in green) around a freshly isolated megakaryocyte (GPIbβ in white). *, sinusoid lumen; arrowheads, basement membrane-cage connections; BM, bone marrow; bm, basement membrane; MK, megakaryocyte; Bars, 10 μm.

The online version of this article includes the following source data and figure supplement(s) for figure 1:

**Figure supplement 1.** Molecular cartography of the ECM around megakaryocytes.

**Figure supplement 1—source data 1.** Source data for *Figure 1—figure supplement 1*.

mechanical dissociation and size exclusion, nearly half of the megakaryocytes successfully retained their cages (53.4 ± 5.6 %, 329 megakaryocytes counted from 3 mice), indicating strong physical attachments between both components (*Figure 1D*). While fibronectin and fibrinogen are present around megakaryocytes and at the vessel-cell interface, they do not form a reticular ECM cage. Furthermore, no connection was found between fibronectin and fibrinogen deposition with the sinusoid basement membrane, in contrast to the findings for laminin and collagen IV (*Figure 1—figure supplement 1K*). These observations demonstrate that megakaryocytes establish precise and tight interactions with an ECM cage made of laminin ϒ1 and collagen IV in a spatially confined microenvironment at the sinusoidal basement membrane.

## Reduced 3D laminin cage and vessel-associated megakaryocytes in Lama4[-/-] mouse bone marrow

Among the γ1 chain-bearing laminin isoforms, laminin α4 was abundantly found in bone marrow sinusoid basement membrane (*Susek et al., 2018*). To explore the direct influence of ECM organization on platelets and megakaryocytes, we utilized laminin α4-deficient mice (*Lama4[-/-]*). These mice exhibit mild thrombocytopenia, with platelet counts approximately 20% lower than wild-type (WT) mice (*Cai et al., 2022*). Their platelets maintain a normal ultrastructure and functions, as shown by flow cytometry and EM analysis (*Figure 2—figure supplement 1A–C*). Bone marrow from *Lama4[-/-]* mice showed a normal number of mature megakaryocytes with characteristic ultrastructural features (*Figure 2C* and *Figure 2—figure supplement 1D*). Analysis of bone marrow explants further demonstrated that these megakaryocytes could extend proplatelets, indicating preserved fundamental function (*Figure 2—figure supplement 1E–F*).

We next examined the ECM cage using an antibody against laminin γ1. In *Lama4[-/-]* mice, laminin γ1 deposition around megakaryocytes and in the sinusoid basement membrane decreased by 1.7- and 2.6-fold compared to controls, indicating a disruption of the laminin cage (*Figure 2A–B*, upper panels). However, collagen IV organization remained unaltered around the megakaryocytes in these mice (*Figure 2A–B*, lower panels), showing that laminin α4 is not necessary for collagen IV

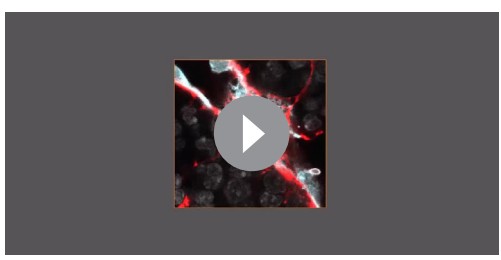

**Video 1.** Video of the laminin γ1 cage around peri-sinusoidal megakaryocyte. Representative maximal video showing the immunostaining of laminin γ1 (red).
https://elifesciences.org/articles/104963/figures#video1

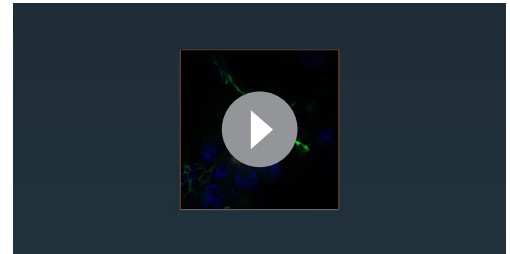

**Video 2.** Video of the collagen IV cage around peri-sinusoidal megakaryocyte. Representative maximal video showing the immunostaining of collagen IV (green).
https://elifesciences.org/articles/104963/figures#video2

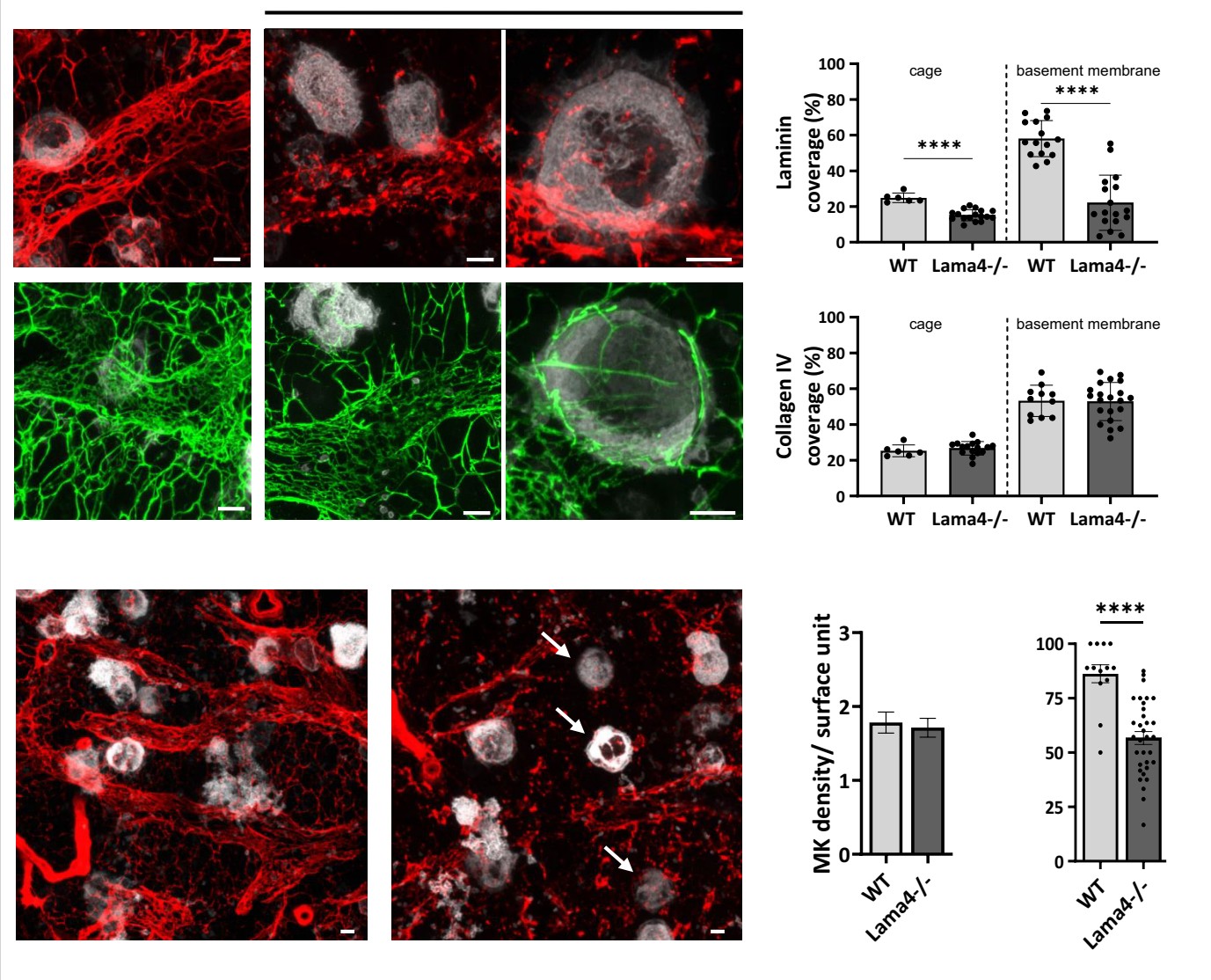

**Figure 2.** Reduced laminin Y1 cage and megakaryocyte-sinusoid interactions in *Lama4-/-* mouse bone marrow. (**A–B**) Depletion of laminin α4 leads to a reduction in the laminin Y1 deposition, but not in collagen IV, in the cage around megakaryocytes and in the sinusoid basement membrane. (**A**) Representative maximal projection images showing the immunostaining of laminin γ1 (red) or collagen IV (green) in the *Lama4-/-* compared to control mice. Two magnifications are shown for *Lama4-/-* mouse. (**B**) Quantification of laminin γ1 and collagen IV surface coverage per megakaryocyte and per basement membrane surface (laminin: 6<cage < 17 and 15<bm < 17; collagen IV: 6<cage < 16 and 11<bm < 21 expressed as a percentage, ****p>0.001 unpaired t-Test). (**C–E**) Depletion of laminin α4 leads to a decrease in the sinusoid-associated megakaryocytes. (**C**) Representative maximal projections showing the immunostaining of laminin γ1 (red) and megakaryocytes (GPIbβ in white) in the bone marrow of *Lama4-/-* and control mice. (**D**) Quantification of the total number of megakaryocytes per surface unit (s.u., 12,945 μm2, n=3 for each genotype, 87<n < 92, p=0.9228, ns, Mann-Whitney test). (**E**) Quantification of the sinusoid-associated megakaryocytes (n=3, ****p>0.001 unpaired t-Test) in control (gray) and *Lama4-/-* (dark) mice. Arrows point to megakaryocytes, which are not associated with sinusoids (pMK). bm, basement membrane; MK, megakaryocyte; pMK, MK in the parenchyma; Bar, 10 μm.

The online version of this article includes the following source data and figure supplement(s) for figure 2:

**Source data 1.** Source data for *Figure 2*.

**Figure supplement 1.** Characterization of Lama4-/- platelets and megakaryocytes.

**Figure supplement 1—source data 1.** Source data for *Figure 2—figure supplement 1*.

cage formation. We also analyzed megakaryocyte localization in the bone marrow of *Lama4*[-/-] mice (*Figure 2C*). The total megakaryocyte density was similar to that of control mice (*Figure 2D*), but we observed more parenchymal MK with a 1.5-fold reduction in the number of megakaryocytes near sinusoids (*Figure 2E*). These observations highlight the role of the laminin α4 cage in maintaining optimal positioning of megakaryocytes near sinusoids.

## Integrins maintain the structural properties of the ECM cages

Integrins play a crucial role in ECM remodeling. Megakaryocytes express β1 and β3 integrins as main ECM receptors (*Yang et al., 2022*). To elucidate the molecular mechanisms governing the intricate interactions between megakaryocytes and the ECM cage, we investigate the subcellular localization of integrins on immunostained ultrathin bone marrow cryosections. Using conformation-independent antibodies, we showed the presence of β1 subunit on the plasma membrane and on the DMS (*Figure 3A*). Similar findings were obtained for total β3 integrins (*Figure 3—figure supplement 1A*). In contrast, the activated form of β1 integrin was expressed only at interaction sites with laminin γ1 and collagen IV, suggesting that activated β1 integrins mediate megakaryocyte interactions with the ECM cage (*Figure 3B*). Activated β3 integrin involvement remains to be determined, as no specific signal was detected when testing two different antibodies (JonA-PE, Pac 1).

To test if integrin-mediated signaling plays a role in the structural assembly of the ECM cage, a transgenic mouse model lacking β1 and β3 integrins in the megakaryocyte lineage was used (*Itgb1*[-/-]/*Itgb3*[-/-]) (*Figure 3C*). *Itgb1*[-/-]/*Itgb3*[-/-] mice were generated by crossing the two single loxP-flanked lines (*Itgb1*[-/-] and *Itgb3*[-/-] mice) with *Pf4-Cre* mice (expressing the Cre recombinase under the control of the *Pf4* promoter). Our analysis revealed a 2.6-fold reduction in laminin γ1 deposition on the megakaryocyte surface in *Itgb1*[-/-]/*Itgb3*[-/-] mice compared to *Pf4-Cre* control mice. Furthermore, intensity profiles demonstrated that laminin γ1 forms a less dense network around the megakaryocytes in the double knockout mice, with significantly increased mesh sizes (10.6±1.3 μm vs 6.3±0.6 μm, respectively; *Figure 3D*). Notably, this lower ECM density surrounding megakaryocytes in *Itgb1*[-/-]/*Itgb3*[-/-] mice is not linked to increased matrix degradation or reduced laminin synthesis (*Figure 3—figure supplement 1B–C*). As expected, the laminin γ1 network at the basement membrane remained unaffected in these mice, as the integrin deletion is restricted to megakaryocytes (*Figure 3—figure supplement 1E–F*).

Further confocal and immunoEM examination revealed a decrease in the expression of fibrillar fibronectin around *Itgb1*[-/-]/*Itgb3*[-/-] megakaryocytes, along with mislocalization of fibrinogen in the extracellular space of the DMS instead of the granules (*Figure 3—figure supplement 1G and H*). These findings show that megakaryocytes, through β1 and β3 integrin-dependent mechanisms, participate in their ECM microenvironment remodeling (ECM cage and fibronectin, fibrinogen deposition) and that megakaryocytes β1 and β3 integrins are required.

## Normal ECM organization is essential for proper megakaryocyte intravasation

Next, we investigate the relevance of the ECM cage to platelet and megakaryocyte functions. Flow cytometry showed that *Itgb1*[-/-]/*Itgb3*[-/-] platelets have intact α-granule secretion but impaired integrin-mediated aggregation, consistent with the bleeding phenotype observed in these mice (*Janus-Bell et al., 2024*; *Figure 4—figure supplement 1A*). As we previously described that *Itgb1*[-/-]/*Itgb3*[-/-] mice had a 50% reduction in platelet count (*Guinard et al., 2023*), we hypothesized that the organization of the ECM cage could contribute to the maturation process of megakaryocytes. Firstly, the number, size, and ploidy of megakaryocytes in *Itgb1*[-/-]/*Itgb3*[-/-] mice are similar to control mice. However, these mutant megakaryocytes show increased emperipolesis and severe DMS dysplasia, characterized by excessive membrane accumulation and abnormal architecture. Typical platelet territories are poorly defined, indicating disrupted maturation processes (*Figure 4—figure supplement 1B–E*). We also analyzed megakaryocyte localization in the bone marrow tissue. We found a significantly higher proportion of megakaryocytes that were extending intravascular fragments (i.e. intravasation) in *Itgb1*[-/-]/*Itgb3*[-/-] mice (15.1 ± 2.7% of total bone marrow megakaryocytes) compared to that in *Pf4Cre* mice (2.9 ± 0.4% of total megakaryocytes). More strikingly, intact megakaryocytes were found in the sinusoid lumen, a rare phenomenon in control mice under physiological conditions (*Figure 4A–B*).

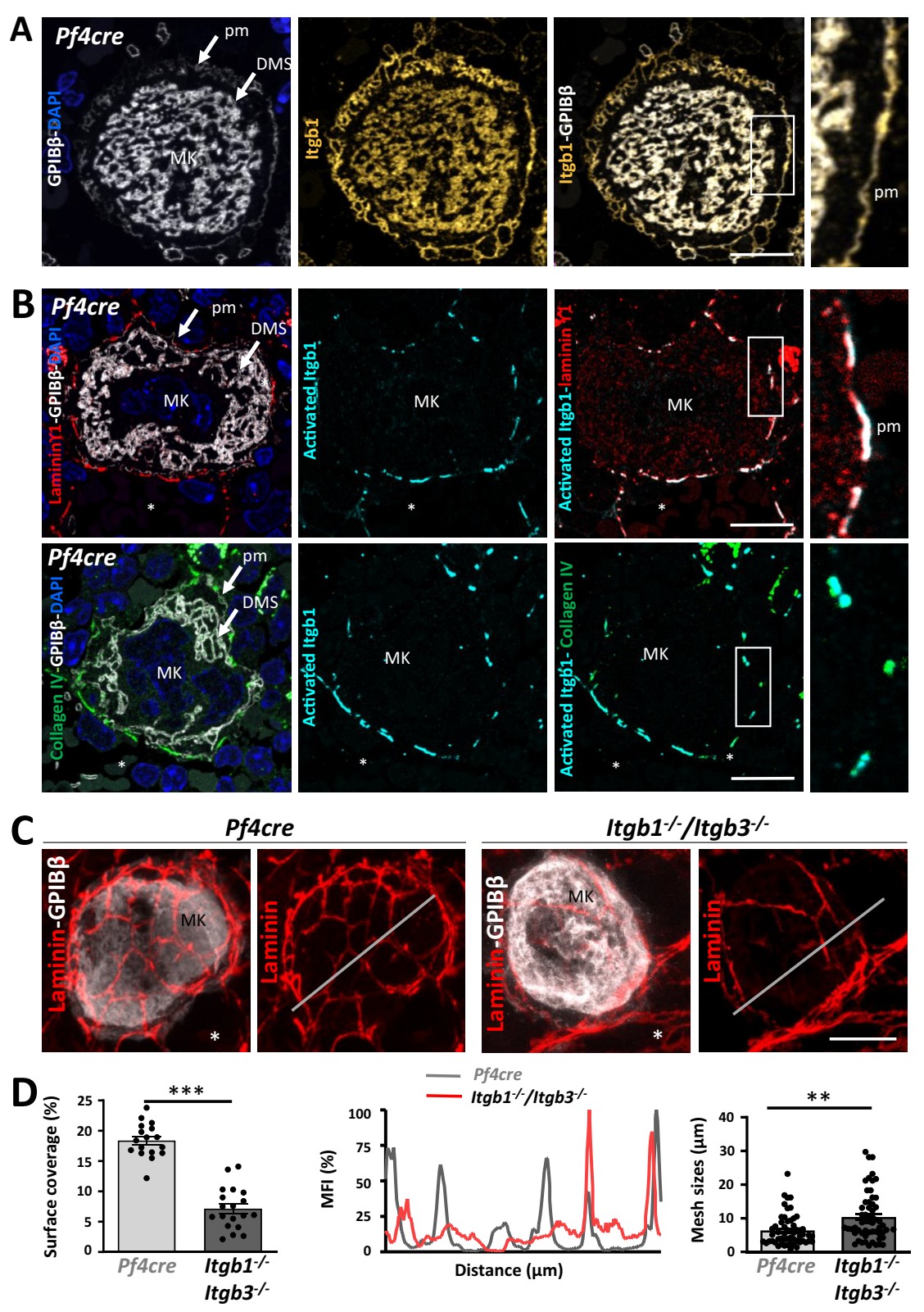

**Figure 3.** Integrin-mediated control of the 3D ECM cage around megakaryocytes. (**A**) Representative 2D images of *Pf4cre* bone marrow cryosections (250 nm) showing a sinusoid-associated megakaryocyte immunostained for β1 integrin (MAB1997 in yellow). Right: the boxed area is shown at a higher magnification. (**B**) Activated β1 integrins form functional adhesion structures around megakaryocyte surfaces. Representative 2D images of *Pf4cre* bone marrow cryosections showing laminin (red, underline{upper panels}) or collagen IV (green, underline{lower panels}) staining around a sinusoid-associated megakaryocyte

*Figure 3 continued on next page*

*Figure 3 continued*

immunostained for activated β1 integrin (9EG7 in cyan) and GPIbβ (white). Right: Magnification of the boxed area showing co-localization of the ECM proteins and activated β1 integrin. (**C**) Depletion of β1 and β3 integrins leads to a reduction in the laminin deposition on the surface of megakaryocytes. Representative 3D images showing a decrease in laminin deposition (red) on *Itgb1-/-/Itgb3-/-* megakaryocytes compared to *Pf4cre*. (**D**) Quantification of laminin surface coverage per megakaryocyte (in %, 17<n < 19 as indicated in the bars, ***p>0.001 unpaired t-Test), (Middle) expression profile of the laminin staining along straight lines (25 µm long) visible as white lines in the confocal images, and (Right) quantification of mesh sizes (in µm, 14<n < 16, **p<0.01 Mann-Whitney). *, sinusoid lumen; MFI, mean fluorescence intensity; MK, megakaryocyte; pm, plasma membrane; Bars, 10 µm.

The online version of this article includes the following source data and figure supplement(s) for figure 3:

**Source data 1.** Source data for *Figure 3*.

**Figure supplement 1.** Integrins control the structural properties of the ECM cages around megakaryocytes.

**Figure supplement 1—source data 1.** Source data for *Figure 3—figure supplement 1*.

To elucidate which of the two integrins was responsible for the observed phenotype, we employed single-knockout mice (*Figure 4C*). Analysis of the ECM cage revealed a trend toward decreased laminin ϒ1 density around *Itgb1⁻/⁻* megakaryocytes, while no notable changes were observed around *Itgb3⁻/⁻* cells. No statistically significant increase in intravasation events was observed in either single knock-out mouse model. This suggests a redundant role of both integrins in regulating the megakaryocyte ECM cage and the intravasation process, potentially compensating for each other's role. Additionally, we investigate the role of GPVI, one of the two well-characterized collagen receptors in this process (*Figure 4C*), and no alterations in ECM cage formation or intravasation behavior were noticed, which is in agreement with previous reports (*Semeniak et al., 2019*).

To understand the unusual localization of circulating *Itgb1⁻/⁻/Itgb3⁻/⁻* megakaryocytes in the bloodstream, we used intravital 2-photon microscopy imaging to describe the dynamic of the megakaryocyte behavior. We used the GPIX marker, a subunit of the GPIb-V-IX complex that is expressed on mature megakaryocytes, to track the behavior of megakaryocytes in living mice. Among the stabilized megakaryocytes at the parasinusoidal interface, we could observe the cellular distortions of *Itgb1⁻/⁻/Itgb3⁻/⁻* megakaryocytes and their exit of the marrow (*Figure 4D–E*, *Video 3*). Intact *Itgb1⁻/⁻/Itgb3⁻/⁻* megakaryocytes were not detected in the peripheral blood by either flow cytometry or blood smear analysis (*Figure 4—figure supplement 1F*), indicating that megakaryocytes do not normally circulate in the systemic bloodstream. Instead, large MK nuclei were found trapped specifically in the lung capillaries (*Figure 4F–G*), consistent with their known physical retention in the pulmonary circulation where they can release platelet. Transmission electron microscopy (TEM) observation confirmed that intact *Itgb1⁻/⁻/Itgb3⁻/⁻* megakaryocytes were similar in size and ultrastructure to those in the stroma compartment (*Figure 4H*). Furthermore, no significant change in the size of the endothelial pores (*Itgb1⁻/⁻/Itgb3⁻/⁻*: 4.6±0.3 µm; *Pf4Cre*: 4.3±0.4 µm) was observed, indicating that increased megakaryocyte intravasation is not linked to endothelium alteration in mutant mice.

## Integrins promote megakaryocyte adhesion to the ECM components of the bone marrow

We next assessed the contribution of integrins' adhesive function in megakaryocyte anchorage to ECM. Under static conditions, most *Itgb1⁻/⁻/Itgb3⁻/⁻* megakaryocytes did not spread and still showed a round shape on immobilized laminin, fibronectin, or fibrinogen (*Figure 5A–B*). We next measured the adhesive potential of freshly isolated bone marrow megakaryocytes to test for such an adhesive role. To this end, we used a miniaturized microfluidic-based experimental model in which individual megakaryocyte detachment was tracked when exposed to a flow rate of 300 s-1, similar to the flow typically found in sinusoids (*Figure 5—figure supplement 1A-B*). *Pf4-Cre* megakaryocytes had capture yields of 70.5% on laminin, 83.6% on fibrillar fibronectin, and 82.0% on fibrinogen. Laminin exhibited the least adhesion, emphasizing the importance of the molecular composition of the said 3D ECM cage and its evident impact on the adhesion response of megakaryocytes in vivo (*Figure 5C–D*). *Pf4-Cre* megakaryocytes remained anchored, while *Itgb1⁻/⁻/Itgb3⁻/⁻* megakaryocytes experienced higher detachment rates across various ECMs (*Videos 4 and 5*). In line with this conclusion, we used the bone marrow explant model to study the adhesion properties of the megakaryocytes associated with their 3D ECM cage (*Figure 5E–F*). This model enabled us to quantify the physical detachment of megakaryocytes at the periphery of the explants. After 3 hr, more megakaryocytes detached from

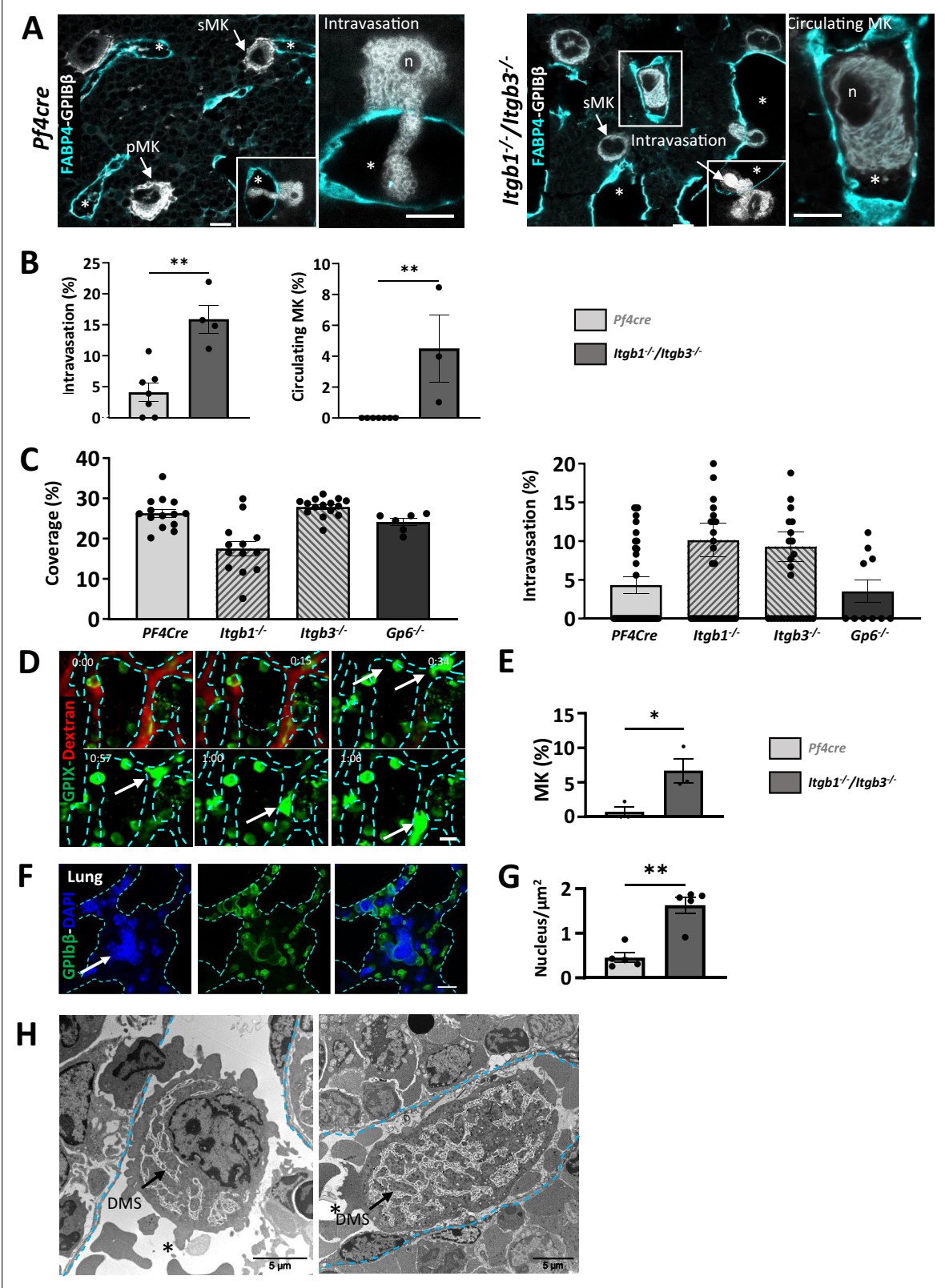

**Figure 4.** Integrins protect megakaryocytes from entering the bloodstream as whole cells (**A–C**) Higher proportion of intravasation events in mice lacking β1/β3 integrins. (**A**) Representative confocal images of *Pf4cre* and *Itgb1-/-/Itgb3-/-* whole-mount bone marrow immunostained for GPIbβ (white) and FABP4 (cyan). (**B**) Quantification of megakaryocyte intravasation and circulating megakaryocytes (3 mice minimum for each genotype, 213<n < 397 for Pf4cre and *Itgb1-/-/Itgb3-/-*, ** p<0.001 Mann Whitney). (**C**) Quantification of the laminin γ1 deposition in the ECM cage in single knockout integrins

*Figure 4 continued on next page*

*Figure 4 continued*

and in Gp6 knockout. Quantification of the intravasation events in single knock-out mice showing that both integrins are essential for the proper anchoring of megakaryocytes in their vascular niche. (**D–E**) Intravital two-photon imaging of *Itgb1-/-/Itgb3-/-* mouse calvarial bone marrow. (**D**) Tissues were stained with intravenously injected AF488-conjugated anti-GPIX antibody and rhodamin dextran. The white arrow indicates an intrasinusoidal *Itgb1-/-/Itgb3-/-* megakaryocyte, dotted lines illustrate the sinusoid wall and the values in the left corner show the time-lapses. (**E**) Quantification of circulating megakaryocytes, expressed as a percentage of the total number of megakaryocyte (from three independent experiments, 130<n < 136, 0.0279 *p<0.1, Paired t-test). (**F–G**) Large megakaryocyte nuclei detected in the pulmonary capillaries of *Itgb1-/-/Itgb3-/-* mice. (**F**) Representative confocal images showing megakaryocytes' nucleus (arrow, GPIbβ green, DAPI nucleus) within the pulmonary microvessels of *Itgb1-/-/Itgb3-/-* mice. Cyan dotted lines indicate the vessel wall. (**G**) Quantification of the intravascular megakaryocytes (from five independent experiments, 28<n < 149, 0.0079 **p<0.01, Mann-Whitney). (**H**) Two TEM images showing intravascular entire *Itgb1-/-/Itgb3-/-* megakaryocyte. *, sinusoid wall; FG, fibrinogen; FN, fibronectin; FG, fibrinogen; n, nucleus; sMK, sinusoid-associated MK; pMK, MK in the parenchyma; PPT, proplatelets; Bars in A-G, 10 μm; Bar in E, 5 μm; Bar in G, 30 μm.

The online version of this article includes the following source data and figure supplement(s) for figure 4:

**Source data 1.** Source data for *Figure 4*.

**Figure supplement 1.** Characterization of *Itgb1-/-/Itgb3-/-* platelets and megakaryocytes.

**Figure supplement 1—source data 1.** Source data for *Figure 4—figure supplement 1*.

---

*Itgb1-/-/Itgb3-/-* bone marrow tissue than *Pf4-Cre* explants, reaching a plateau at 6 hr. These findings indicate that β1 and β3 integrins control ECM-megakaryocyte interactions in their native bone marrow microenvironment.

Collectively, our results highlight the essential roles of β1 and β3 integrins in forming 3D ECM cages around megakaryocytes and modulating their adhesion within the bone marrow, which helps stabilize the cells in their vascular niche and prevent the passage of intact megakaryocytes through the sinusoid barriers.

## Cage compression via metalloproteinase inhibition affects the maturation of megakaryocytes

Our results suggest that a weakened ECM cage promotes megakaryocyte intravasation through reduced adhesion. We, therefore, tested whether an increase in cage density also may affect megakaryocyte functions. For that purpose, we proposed to decrease the catabolic aspect occurring in the ECM equilibrium between its synthesis and degradation. To this aim, mice were injected with batimastat (30 mg/kg) and ilomastat (20 mg/kg) for 7 days, in order to inhibit the activation of MMPs in the megakaryocyte microenvironment (*Winer et al., 2018*; *Figure 6—figure supplement 1A*). These mice produced platelets normally in terms of both count and average platelet function (*Figure 6—figure supplement 1B–C*). Regarding the cage, this treatment increased the ECM density in megakaryocyte surrounding, as evidenced by reduced fiber length and pore size (*Figure 6A–B*).

We investigated whether this denser ECM cage might affect megakaryopoiesis. Remarkably, treatment with MMP inhibitors resulted in a significantly higher proportion of smaller megakaryocytes compared to untreated mice (white arrows in *Figure 6C–D*). To better evaluate the extent of this defect, we employed TEM (*Figure 6E*). We found that 53.7% of the megakaryocytes lacked the appropriate organization

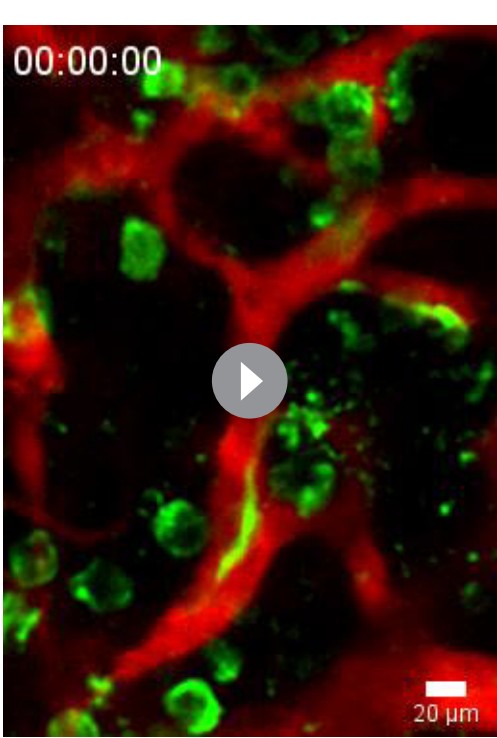

**Video 3.** Intravital two-photon imaging of *Itgb1-/-/Itgb3-/-* mouse calvarial bone marrow. Tissues were stained with intravenously injected AF488-conjugated anti-GPIX antibody and rhodamine dextran.
https://elifesciences.org/articles/104963/figures#video3

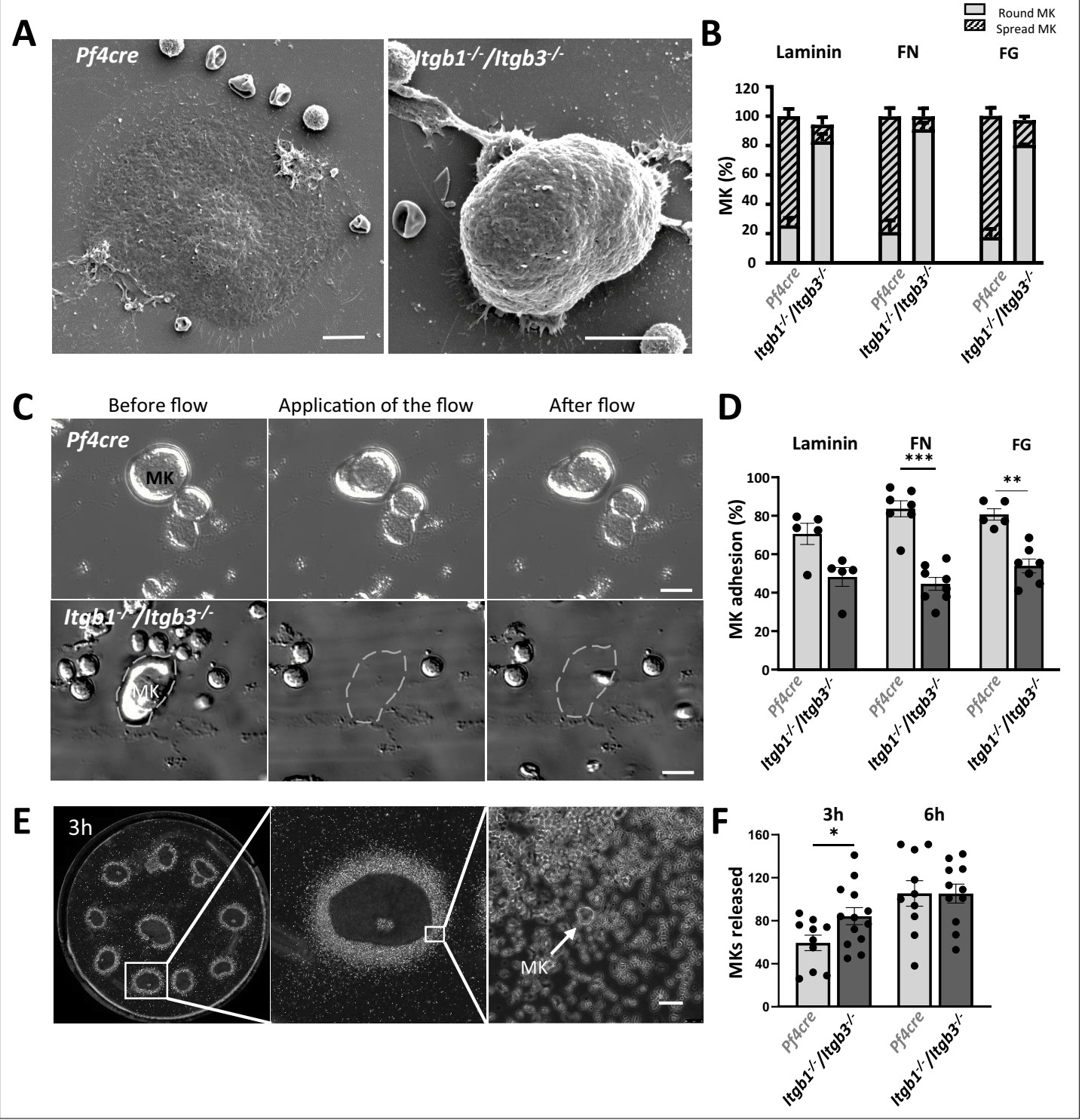

**Figure 5.** Integrins promote megakaryocyte adhesion to the ECM components of the bone marrow. (**A–B**) Impaired adhesion and spreading of *Itgb1-/-/Itgb3-/-* megakaryocytes. (**A**) Representative SEM images depicting bone marrow-derived megakaryocytes adhering on laminin. (**B**) Spreading (hatched bars) and round (gray bars) megakaryocytes were counted following 3 hr incubation on laminin, fibronectin (FN), and fibrinogen (FG) (in %) (from four to six independent experiments). (**C–D**) Microfluidic flow chamber to study megakaryocyte adhesion efficiency. (**C**) Representatives bright field images showing that upon flow application, *Itgb1-/-/Itgb3-/-* megakaryocytes detach from fibrillary fibronectin protein, while *Pf4Cre* MKs remain attached. (**D**) Quantification of the detachment of *Pf4cre* and *Itgb1-/-/Itgb3-/-* megakaryocytes on laminin, fibrillar fibronectin, and fibrinogen (from five to seven independent experiments,**p<0.01, ***p<0.001, Mann-Whitney). (**E–F**) Reduced physical anchoring of *Itgb1-/-/Itgb3-/-* megakaryocytes to bone marrow. (**E**) Representatives bright field images of the ten femur bone marrow sections placed in an incubation chamber (left panel), of the box (center

*Figure 5 continued on next page*

*Figure 5 continued*

panel) and of the megakaryocytes released from the periphery of the explants (right panel). (F) Quantification of the number of *Pf4cre* and *Itgb1-/-/Itgb3-/-* megakaryocytes released from the explants following 3 hr (from 10 to 13 independent experiments, 594<n < 1095 for Pf4cre and *Itgb1-/-/Itgb3-/-*, *p<0.05, unpaired t-test). Dotted lines, MK detachment; MK, megakaryocytes, FN, fibronectin; FG, fibrinogen; n, number of cells studied; Bars in A, 10 µm; Bars in B, 20 µm; Bars in B, 30 µm.

The online version of this article includes the following source data and figure supplement(s) for figure 5:

**Source data 1.** Source data for *Figure 5*.

**Figure supplement 1.** Adhesive properties o fItgb1-/-/Itgb3-/- megakaryocytes.

**Figure supplement 1—source data 1.** Source data for *Figure 5—figure supplement 1*.

of the DMS in MMP inhibitors-treated mice, reflecting a primary failure in the cytoplasmic maturation (Stage II in *Figure 6F* right). These mice also showed a significant increase in the total megakaryocyte number compared to the control condition (*Figure 6D and F* left), consistent with altered bone marrow ECM remodeling affecting megakaryocyte retention. To assess whether MMP inhibition directly affects megakaryocyte maturation and density, we cultured primary megakaryocytes in the presence of increasing concentrations of inhibitors. We observed no significant change in proliferation, maturation, or proplatelet formation (*Figure 6—figure supplement 1D*). While this in vitro model and the application of exogenous MMP inhibitors offer valuable mechanistic insights, they do not fully recapitulate the complexity and regulation of the bone marrow ECM in vivo. For example, MMPs from other cell types and paracrine signals in the bone marrow may influence megakaryocyte behavior. Nonetheless, these findings are consistent with the conclusion that disrupting ECM remodeling significantly impacts megakaryocyte maturation within the physiological bone marrow.

Immunofluorescence analysis of bone marrow sections revealed an upregulation of activated β1 integrin (*Figure 6—figure supplement 1E*). These results indicate that increased ECM density may promote megakaryocyte adhesion within the congested microenvironment of the bone marrow. To further investigate this point, we quantified the number of megakaryocytes released from bone marrow explants. We observed a significant reduction in megakaryocyte egress from the bone marrow microenvironment after 6 hr in MMP inhibitor-treated compared to control mice (*Figure 6G–H*).

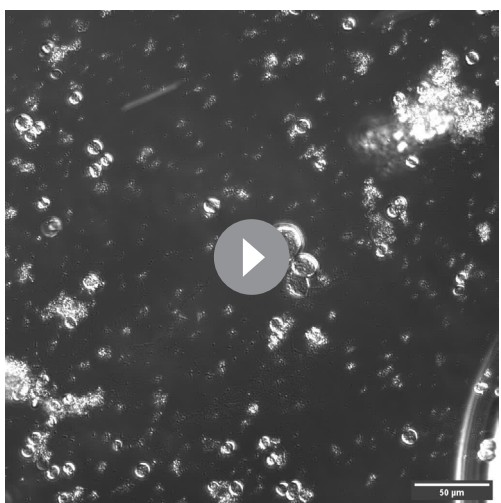

**Video 4.** Microfluidic flow chamber to study Pf4Cre megakaryocyte adhesion efficiency. Representatives bright field video showing that upon flow application *Pf4Cre* MKs remain attached.

https://elifesciences.org/articles/104963/figures#video4

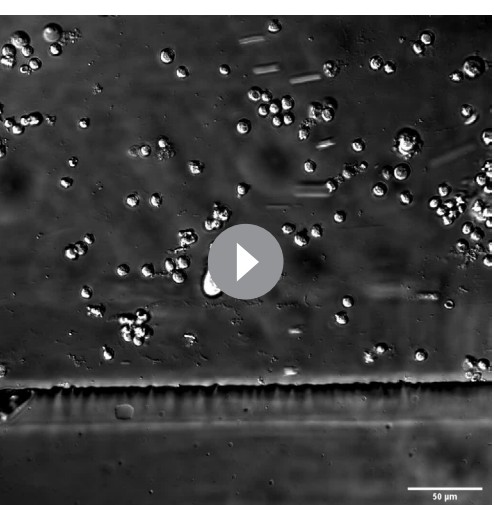

**Video 5.** Microfluidic flow chamber to study Itgb1-/-/Itgb3-/- megakaryocyte adhesion efficiency. Representative bright field video showing that upon flow application, *Itgb1-/-/Itgb3-/-* megakaryocytes detach from fibrillary fibronectin protein.

https://elifesciences.org/articles/104963/figures#video5

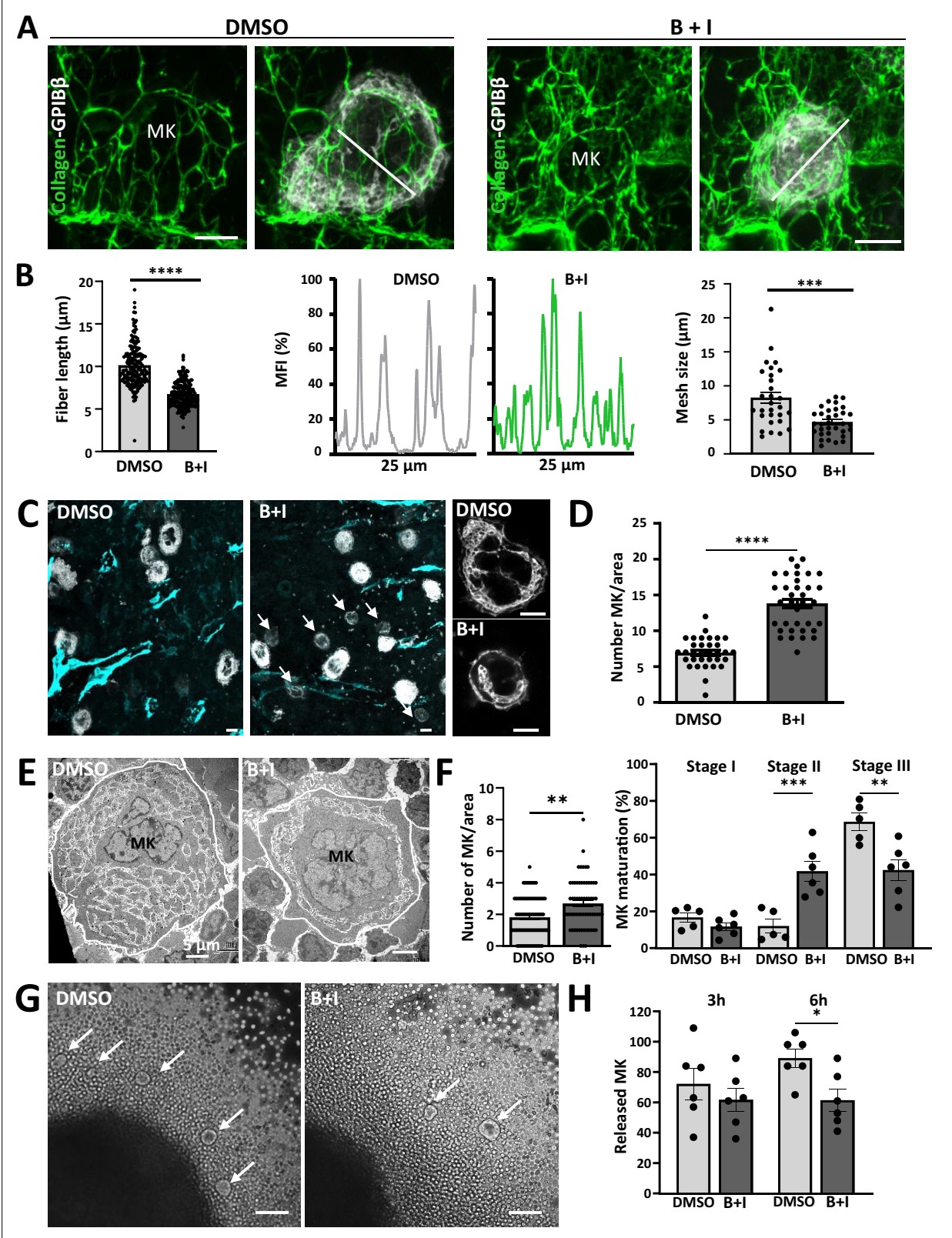

**Figure 6.** Maturation of the 3D ECM cage is correlated with maturation of the DMS in megakaryocytes. (**A–B**) MMP inhibition leads to a densification of the 3D ECM cage. (**A**) Representative 3D confocal images showing a significant increase in collagen IV deposition (green) on the megakaryocyte surface in treated mice treated with the intravenous cocktail of protease inhibitors (B+I). (**B**) (Left) Quantification of collagen IV fluorescence showed a shortening of collagen IV fibers in treated mice. compared to that in control mice (from three to five independent experiments, 20<n < 22 as indicated

*Figure 6 continued on next page*

*Figure 6 continued*

in the bars, ****p<0.001, Mann-Whitney), (Middle) Histograms of fluorescence intensity versus distance showed an increase in cross-linking with a reduction in pore size (white lines of 25 μm length are visible in the confocal images), (Right) Reduction in mesh sizes in treated mice (from three independent experiments, 7<n < 12, ***p>0.001, t-test). (**C–D**) MMP inhibition affects megakaryocyte growth. (**C**) Representative confocal images from DMSO *vs* B+I treated mice immunostained for GPIbβ (white) and FABP4 (cyan), zoomed-in images showing the difference in megakaryocyte size between the two groups (arrows). (**D**) Quantification of the number of megakaryocytes per bone marrow area (194x194 μm; from three independent experiments, 229<n < 483 as indicated in the bars, ****p<0.0001, Mann-Whitney). (**E–F**) MMP inhibition leads to an increase in the number of immature megakaryocytes. (**E**) TEM observation revealed the presence of numerous immature megakaryocytes (stage II) in treated mice as compared to fully mature megakaryocytes in control mice (stage III). (**F**) Quantification of the total number of megakaryocytes (145<n < 169 as indicated in bars, **p<0.01, Mann-Whitney) and in the proportion of immature megakaryocytes (stage II) in the B+I group (from three independent experiments, **p<0.05, ***p<0.01, one-way ANOVA with Tukey correction). (**G–H**) MMP inhibition reduced release of megakaryocytes from the bone marrow explant. (**G**) Representative bright field images of the megakaryocytes (arrows) released from the periphery of the control and treated explants. (**H**) Quantification of the number of megakaryocytes released following 3 hr and 6 hr (from six independent experiments, 369<n < 783 for DMSO and B+I, *p<0.05, unpaired t-test). B+I, batimastat + Ilomastat; sMK, sinusoid-associated MK; pMK, MK in the parenchyma; n, number of cells studied; Bars in A and C, 10 μm; Bars in E, 5 μm; Bar in G, 50 μm.

The online version of this article includes the following source data and figure supplement(s) for figure 6:

**Source data 1.** Source data for *Figure 6*.

**Figure supplement 1.** Proteolysis inhibition does not affect platelet count and in vitro megakaryocyte maturation.

**Figure supplement 1—source data 1.** Source data for *Figure 6—figure supplement 1*.

Together, these results indicate that MMPs are essential regulators of ECM homeostasis within the bone marrow, influencing megakaryocyte maturation and attachment to the vascular niche.

## Discussion

Physiological platelet formation relies on the strategic positioning of mature megakaryocytes at sinusoids to facilitate polarized proplatelet extension and platelet release into circulation. This study identifies a 3D ECM cage anchoring megakaryocytes to bone marrow sinusoids. This cage, composed of laminin γ1 and α4 and collagen IV fibers, extends from the basement membrane and surrounds

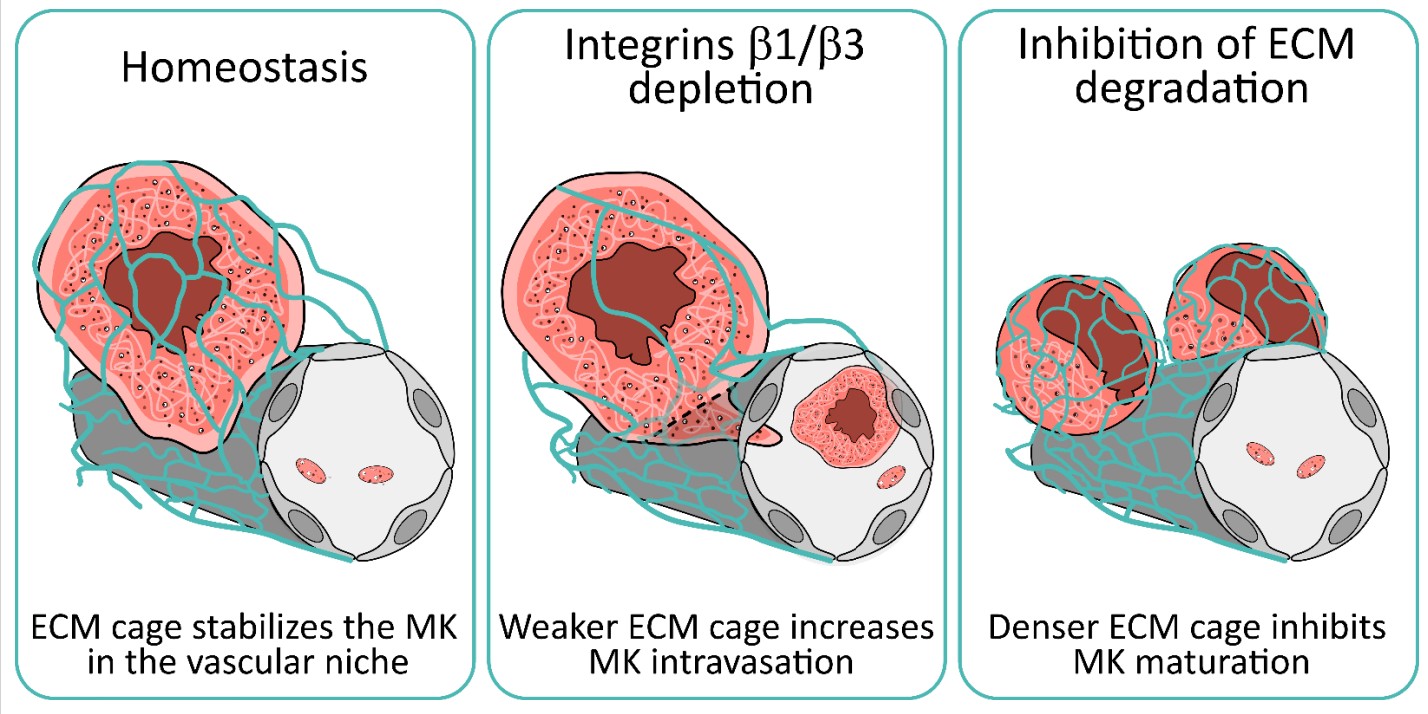

**Figure 7.** Integrin-mediated signaling and MMP proteolysis regulate the matrix remodeling and the adhesive properties of the 3D ECM cage, which control megakaryocyte maturation and intravasation at the bone marrow-blood interface.

megakaryocytes. Megakaryocyte integrin signaling and ECM proteolysis regulate its remodeling and adhesive properties. This dynamic microenvironment is key for megakaryocyte positioning, maturation, and intravasation, representing a novel concept in understanding physiological platelet formation (Graphical Abstract, *Figure 7*).

In line with previous studies (*Guinard et al., 2023*; *Larson and Watson, 2006*; *Malara et al., 2014*; *Semeniak et al., 2016*), we identified the presence of collagen IV, laminin γ1, fibronectin, and fibrinogen surrounding sinusoid-associated megakaryocytes. Notably, when analyzed using immunofluorescence and transmission electron microscopy (TEM), collagen I fibers were absent under steady-state conditions. The discrepancy between our results and those reported in the literature may be attributed to differences in staining methodologies or variations in the physiological context of the bone marrow. Furthermore, our findings indicate that GPVI-deficient megakaryocytes do not display alterations in ECM cage formation, vessel association, and intravasation behavior. These observations support studies showing GPVI and α2β1 integrins are not critical for megakaryocyte function, underscoring potential redundancy in ECM-receptor interactions (*Semeniak et al., 2019*). Besides, non-protein ECM molecules, including glycosaminoglycans, have been shown to play an essential role in supporting megakaryocyte function, including maturation (*Petrey et al., 2016*; *Piszczatowski et al., 2022*; *Vögtle et al., 2019*).

We identify laminin and collagen IV as the principal structural components of this ECM cage. Consistent with previous research (*Abbonante et al., 2016*; *Cai et al., 2022*; *Susek et al., 2018*), these ECM proteins were detected intracellularly within megakaryocytes and in stromal cells situated in their vicinity. This indicates that the ECM cage proteins may be derived from autocrine synthesis and their production by neighboring cells. Interestingly, while fibronectin and fibrinogen were found around megakaryocytes, they do not form an ECM cage, suggesting distinct functional roles, potentially related to megakaryocyte expansion in bone marrow fibrosis (*Malara et al., 2019*; *Matsuura et al., 2020*).

Our findings in *Lama4*-deficient mice reveal that the ECM cage is critical for positioning megakaryocytes near sinusoids. It is established that CXCL12 is a most potent physiological chemoattractant for megakaryocytes and that the parasinusoidal location of megakaryocytes in the bone marrow is mediated by the CXCL12/CXCR4 interaction (*Hamada et al., 1998*; *Wang et al., 2019*). Other chemoattractive factors also include fibroblast growth factor 4 (FGF4; *Avecilla et al., 2004*) and VWF (*Ouzegdouh et al., 2018*). Besides its potential role as a reservoir of such growth factors, we show that the ECM cage is also a crucial element in maintaining megakaryocytes in the proximity of sinusoids, which is essential for their efficient maturation and optimal platelet production. We also find the persistent presence of this ECM cage throughout megakaryocyte maturation, supporting previous findings that megakaryocyte development occurs primarily at the sinusoid (*Gaertner et al., 2024*; *Lichtman et al., 1978*; *Stegner et al., 2017*). Together, these findings underscore the essential structural and functional role of the ECM cage in ensuring megakaryocyte localization at the sinusoid niche, which is indispensable for their proper maturation. However, we also acknowledge that intrinsic cellular defects may contribute to maturation impairments observed in laminin and integrin-deficient models.

Our study highlights the indispensable roles of β1 and β3 integrins in maintaining megakaryocyte stability within the vascular niche. We demonstrate that these integrins are crucial for two fundamental processes: (1) the building of the 3D ECM cages and (2) the ECM-mediated adhesion mechanisms to the vascular niche. Integrins' biochemical and mechanical signal transduction play key roles in ECM remodeling (*Larsen et al., 2006*) and can generate the forces that may cause these changes (*Lemmon et al., 2009*). For instance, RhoA inhibition in *Itgb1⁻/⁻/Itgb3⁻/⁻* megakaryocytes (*Guinard et al., 2023*) is likely contributing to the altered ECM remodeling. Nevertheless, we still need to understand how exactly β1 or β3 integrins participate in ECM remodeling and ECM cage dynamics. Our work further demonstrates that in these mice, intact megakaryocytes can cross the sinusoidal vessel wall. Of note, no intravascular megakaryocytes were found in *Lama4*-deficient mice, which have a normal collagen IV cage but a compromised laminin cage. This indicates that the integrity of all components of the cage is not needed to keep megakaryocytes from entering circulation. Importantly, circulating megakaryocytes have also been observed in human lungs during emergency megakaryopoiesis, such as in cases of infectious diseases, inflammatory conditions, and following cardiopulmonary bypass procedures (*Frydman et al., 2023*; *Gelon et al., 2022*; *Puhm et al., 2023*; *Rapkiewicz et al., 2020*).

These circulating megakaryocytes have been implicated in thrombotic complications, underscoring the importance of integrin signaling-dependent mechanisms regulating megakaryocyte retention and release.

The precise role of the ECM cage in platelet production remains incompletely understood. In the Lama4⁻/⁻ mice, a collagen-rich ECM cage persists with normal fibronectin deposition, whereas the *Itgb1⁻/⁻/Itgb3⁻/⁻* model displays a much more severe phenotype, characterized by the loss of both the laminin cage and collagen, along with the absence of fibrillar fibronectin. The preserved collagen and fibronectin in Lama4⁻/⁻ mice may allow for residual activation of signaling pathways - potentially through integrins or alternative mechanisms - compared to the *Itgb1⁻/⁻/Itgb3⁻/*model, where these matrix components are absent.

MMPs profoundly influence megakaryocyte maturation and their association with the vascular niche by modulating the density of ECM cage architecture. This finding highlights the dynamic nature of the bone marrow microenvironment and its impact on megakaryopoiesis. Dysregulation of ECM production, often associated with bone marrow pathologies, can lead to its uncontrolled accumulation. In myelofibrosis, for instance, megakaryocytes secrete transforming growth factor-β1 (TGF-β1) which stimulates excessive ECM production by stromal cells (*Abbonante et al., 2016*). This pathological ECM accumulation leads to abnormally smaller megakaryocytes that exhibit reduced platelet production (*Gianelli et al., 2023*; *Malara et al., 2018*; *Sarachakov et al., 2023*). Likewise, we observed smaller and immature megakaryocytes retained in the constrained ECM microenvironment of the bone marrow, associated with an upregulation of activated β1 integrin in treated mice. Interestingly, previous research by K. Hoffmeister's group has demonstrated that proper megakaryocyte localization at the sinusoids depends on β1 integrin glycosylation. Specifically, the absence of galactosylation in β4galt1⁻/⁻ megakaryocytes leads to hyperactivity of β1 integrin, which negatively impacts the formation of the DMS (*Giannini et al., 2020*). Building on these insights, our study shows the coordinated interplay between integrin-mediated signaling and MMP proteolysis, working together to tightly regulate the density and cross-linking properties of the ECM cage and thereby control megakaryocyte maturation at the bone marrow-blood interface. Intriguingly, despite these effects on megakaryocyte maturation, we observed no significant changes in circulating platelet counts, and importantly, platelet function remained preserved. This supports the idea that differences in ECM composition can influence the signaling environment and megakaryocyte maturation, but do not fully abrogate platelet function.

The influence of ECM features - like fiber length and pore size - on megakaryocyte biology is complex and not yet fully understood. Longer ECM fibers may help cells adhere better (*Barriga et al., 2018*; *Dolega et al., 2021*). Larger pores could make it easier for megakaryocytes to mature and to extend proplatelets at the bone marrow-blood interface. Also, the mechanical properties of the ECM, particularly its stiffness, have emerged as critical environmental determinants of megakaryocyte development and function (*Abbonante et al., 2017*; *Aguilar et al., 2016*; *Guinard et al., 2023*; *Leiva et al., 2018*). Notably, megakaryocytes possess mechanosensing capabilities, primarily mediated by the β3 integrin subunit, enabling them to detect and respond to changes in substrate stiffness (*Guinard et al., 2023*). The assembly of extracellular matrix (ECM) fibronectin around megakaryopoiesis involves a more complex interplay of integrins, with both β1 and β3 integrins playing essential roles in fibrillogenesis (*Abbonante et al., 2024*; *Guinard et al., 2023*). This cooperative relationship appears to be conserved across various cell types (*Attieh et al., 2017*; *Mets et al., 2019*; *Kyumurkov et al., 2023*). This may explain why the deletion of both integrins is required to affect the 3D ECM cage and megakaryocyte behavior significantly. We cannot exclude that other ECM ligands of β3 integrins, such as fibronectin, may contribute to the organization of collagen IV and laminin, influencing the overall architecture of the vascular niche (*Petito and Gresele, 2024*; *Yang et al., 2022*; *Zeng et al., 2018*). Indeed, it is known that the fibronectin matrix favors the deposition of other ECM proteins, such as collagens IV and various other glycoproteins (*Saunders and Schwarzbauer, 2019*). This hypothesis underscores the need for further investigation into the spatio-temporal dynamics of ECM component assembly in the megakaryocyte vascular niche.

Overall, our findings reveal the supportive role of the ECM cage in platelet biogenesis. Integrin-mediated ECM interactions are clearly crucial, as demonstrated by the 50% reduction in platelet counts observed in integrin double knockout mice. In summary, the ECM cage acts as a finely tuned

facilitator, optimizing the efficiency and precision of platelet production by guiding megakaryocytes to the right place at the right time.

## Materials and methods
### Animals

We used wild type (WT) mice (C57BL/6 J from C. River, L'Arbesle, France), *Itgb1⁻/⁻/Itgb3⁻/⁻* double knockout mice and *Pf4cre* mice aged 10–15 weeks. *Pf4cre* mice expressed the Cre recombinase under the control of the *Pf4* promoter. The *Itgb1⁻/⁻* (*Potocnik et al., 2000*) and *Itgb3⁻/⁻* (*Morgan et al., 2010*) mice were crossed with mice expressing the Cre recombinase under the control of the Pf4 promoter to obtain inactivation in the MK lineage. *Itgb1⁻/⁻/Itgb3⁻/⁻* double knockout (KO) mice were crosses of the two single KO lines, as described in *Guinard et al., 2023*. *Gp6⁻/⁻* mice and *Lama4⁻/⁻* mice were from Nieswandt and Qian labs (*Bender et al., 2011*; *Cai et al., 2022*). All animal studies were approved by the French legislation for animal experimentation and in accordance with the guide for the care and use of laboratory animals as defined by the European laws (Animal Facility Agreement C-67-482-10).

**Table 1.** Immunofluorescence antibodies.

| Antigen | Origin and isotype | Reference and supplier | Cryosections | Whole-mount |
|---|---|---|---|---|
| **Primary antibody** | | | | |
| Collagen I | | Chemicon (Merck) ab765p | 5 µg/ml | |
| Collagen III | | Invitrogen PA5-34787 | 10 µg/ml | |
| Collagen IV | Rabbit polyclonal | Chemicon (Merck), ab756p | 5 µg/ml | |
| FABP4 | Goat polyclonal | R&D Systems AF1443 | 0.5 µg/ml | |
| Fibrinogen | | Dako (Glostrup, Denmark), A0080 | 10 µg/ml | |
| Fibronectin | Rabbit polyclonal | ab2413 (Abcam; Paris, France) | 10 µg/ml | |
| GPIbβ | | In house (Strasbourg, France) | 1/300 | 1/300 |
| Activated β1 integrin (9EG7) | Rat monoclonal | 553715 (BD Pharmingen) | 10 µg/ml | |
| Pan-laminin | | L9393 (Sigma-Aldrich) | 10 µg/ml | |
| Von Willebrand Factor | Rabbit polyclonal | Dako, A0082 | 10 µg/ml | |
| β1 integrin (MB1.2) | Rat monoclonal | Merck, MAB1997 | 10 µg/ml | |
| LucA5 | Rat IgG2A monoclonal | Emfret analytics, LucA5 | 10 µg/ml | |
| **Secondary antibody** | | | | |
| AF488-anti-rabbit immunoglobulin | Donkey | A21206 (Invitrogen) | | |
| AF488-conjugated anti-rabbit immunoglobulin | Goat | A11070 (Invitrogen) | | |
| AF647-conjugated anti-rabbit immunoglobulin | Goat | A21245 (Invitrogen) | | |
| AF647-conjugated anti-rat immunoglobulin | Goat | A21247 (Invitrogen) | | |
| AF555-conjugated anti-goat immunoglobulin | Donkey | A21432 (Invitrogen) | 1/150 | |
| AF647-conjugated anti-pan-laminin | Rabbit | In house | 25 µg/ml | |
| AF555-conjugated anti-collagen IV | Rabbit | In house | 30 µg/ml | |
| AF647-conjugated anti-GPIX antibody | Rat | nXiaB4 Emfret | | |

**Table 2.** Antibodies used in flow cytometry.

| Antigen | Origin and isotype | Reference and supplier | Fluorophore | Flow cytometry | Whole-mount |
|---|---|---|---|---|---|
| Conjugated antibody | | | | | |
| JONA Activated integrin αIIbβ3 | Rat IgG2b | Emfret analytics M023-2 | PE | Dilution of 1/5 | |
| P-selectin (CD62P) | Mouse IgG2a, κ | BioLegend 148303 | APC | 1 μg/ml | |
| CD45 (clone 30-F11) | Rat IgG2b, κ | eBiosciences 12-0451-82 | PE-A | 1 μg/ml | |
| TER119 | Rat / IgG2b, κ | eBiosciences 17-5921-82 | APC-A | 1 μg/ml | |
| GPIbβ | Rat monoclonal | In house (Strasbourg, France) | A647, A568, A488 | 1 μg/ml | |
| Hoechst-33342 | | Invitrogen H3570 | | | 10 μg/ml |
| Integrin αIIb (CD41 – clone MWReg30) | Rat IgG1 κ | eBiosciences 25-0411-82 | PE-Cy7 | 1 μg/ml | |

## Chemicals

Dimethyl sulfoxide (2438), bovine serum albumin (900.011), saponin (47036), Triton X-100 (T8787), fibrinogen and batimastat (SML0041) (Sigma-Aldrich, Rueil-Malmaison, France), Ilomastat (GM6001, HY-15768) (MedChemExpress, Clinisciences, France), Dulbecco's modified Eagle's medium (DMEM), penicillin, streptomycin, and glutamine (Invitrogen, Cergy-Pontoise, France), Gelatin-Oregon Green 488 conjugate (G-13186) (Life Technologies), Laminin 511 recombinant (Biolamina) and fibronectin from human plasma (341635, Calbiochem) were used in this study.

## Antibodies

See *Tables 1 and 2* for details.

GPIX and GPIbβ are components of the GPIb-IX complex, identifying mature megakaryocytes (*Lepage et al., 2000*). The choice of marker used to identify megakaryocytes in different experiments is primarily based on technical considerations. Intravital experiments have been standardized using AF488-conjugated anti-GPIX to consistently identify mature megakaryocytes. The rest of the manuscript uses GPIbβ (GP1bβ) due to its strong, specific, bright staining.

## Extrusion and preservation of murine bone marrow

Bone marrow extrusion, without leaving any residual bone, was obtained by carefully flushing the femurs of mice with PBS using a 21-gauge needle attached to a 10 ml syringe (see photo in *Figure 1D*; *Scandola et al., 2021*). To maximize the preservation of the bone marrow integrity, we used a double fixation procedure (4% paraformaldehyde and 0.4% glutaraldehyde for 1 hr) immediately after bone marrow extrusion, followed by embedding it in 4% low-melting-point agarose to preserve as much as possible their three-dimensional architecture.

## Isolation of freshly isolated bone-marrow-derived megakaryocytes

Bone marrow megakaryocytes were dissociated using a 21 G needle and filtered through a 40 µm Millipore filter. The filter was then rinsed in DMEM +1% fetal bovine serum, and the isolated megakaryocytes were adjusted to 300 cells/ml in the same medium.

## 2D confocal microscopy on bone marrow cryosections

This study explores the interactions between megakaryocytes and their immediate ECM microenvironment. Ultrathin cryosections were used for their superior axial resolution, offering a two- to threefold improvement over conventional confocal microscopy, which facilitates analysis of signal

superposition. Cryosections from WT extruded bone marrow were used to study the ECM composition and distribution of sinusoid-associated megakaryocytes. Bone marrow was fixed in a mix of 2% paraformaldehyde-0.2 % glutaraldehyde in 0.2 M sodium cacodylate buffer for 1 hr. The fixed samples were infiltrated with 2.3 M sucrose and frozen in liquid nitrogen. Ultrathin cryosections of 250 nm were obtained at −110 °C with a LEICA Ultracut UCT cryo-ultramicrotome (Leica Microsystems). For immunofluorescence staining, cryosections were labeled with primary antibodies and conjugated-secondary antibodies of the appropriate species and DAPI, as reported in *Table 1*. They were examined under a confocal microscope (TCS SP8, Leica) using the 63 x objective with a numerical zoom of 4 (pixel size: 0.09 μm). The bone marrow specimens from three mice were examined under identical conditions, using constant exposure and the same irrelevant antibodies. No fluorescence was detected using isotype-specific control IgG.

## 3D confocal analysis on bone marrow vibratome sections

Whole-mount bone marrow preparations were used to investigate (i) the localization of megakaryocytes in the bone marrow and the lung and (ii) the spatial organization of ECM around sinusoid-associated megakaryocytes. Megakaryocytes were classified based on their maturation stage: stage I (presence of granules, no clear DMS visible), stage II (developing DMS not yet organized), and stage III (DMS organized in platelet territories). Mice from each genotype were analyzed and image acquisitions were performed in a blinded manner.

For megakaryocyte localization, sections of femur bone marrow or lungs fixed in 2% paraformaldehyde and 0.2% glutaraldehyde for 1 hr, with a thickness of 250 μm, were incubated with FABP4 overnight. This was followed by overnight staining with Alexa 568-conjugated anti-GPIbβ antibody and Alexa 488 Donkey anti-Goat for FABP4. Finally, Hoechst 33342 was used for counterstaining for 10 min. The fluorescently labeled tissue was placed cut-face down into incubation chambers and mounted with Mowiol mounting solution. The Leica SP8 confocal microscope was used to collect a series of x-y-z images. The images were typically 194x194 μm x-y size and were collected along the z-axis at 1 μm step size through 50 μm of bone marrow tissue. The 40 x objective with a numerical zoom of 4 (pixel size: 0.142 μm) was used. The lung was imaged using an x63 objective with a 2.5 digital zoom (pixel size: 0.145 μm).

To perform 3D ECM analysis, bone marrow samples were fixed in 4% PFA and 0.4% GA for 1 hr, then embedded in 4% agarose and cut into 250 μm-thick sections using a vibratome. These sections were then incubated overnight at 4 °C with primary antibodies targeting laminin, type IV collagen, fibronectin, and fibrinogen, followed by corresponding secondary antibodies and Hoechst 33342 counterstaining. Series of x-y-z images of typically 46.13*46.13 μm x-y size were collected along the z-axis at 1 μm step size through 15–35 μm of sinusoid-associated megakaryocytes, using the 63 x objective with a numerical zoom of 4 (pixel size: 0.09) from a Leica SP8 confocal microscope. For quantitative spatial analysis of ECM around megakaryocytes, the observations have been made on z-stacks as described by *Voisin et al., 2010. The fluorescence was delineated on the maximum z-stack projection of half a megakaryocyte using Image J software.*

Image processing to perform quantitative analyses of fluorescence profiles and fiber length is explained in *Figure 8*. The random length measurement method uses random sampling to provide unbiased data on laminin/collagen fibers in a 3D cage. Measurements included intervals between different branching points throughout the cage, including branch ends. Processing involved five steps: (1) acquiring 3D images, (2) projecting onto 2D planar sections, (3) selecting random intersection points for measurement, (4) measuring intervals using ImageJ software, and (5) repeating the process for a representative dataset. Briefly, a binary threshold mask (0 for the background and 1 for the ECM network, threshold to reduce signal noise) was generated from the channel showing labeled collagen or laminin. The mask was then applied to the channel showing megakaryocytes, resulting in an image that corresponded to only an active ECM signal that colocalized with megakaryocytes. The mask was then applied to the collagen/laminin channel to determine the amount of fluorescent ECM within the mask. This enabled the determination of the cell area and the elimination of the signal outside the mask.

## Intravital imaging

*Pf4cre* and *Itgb1⁻/⁻/Itgb3⁻/⁻* mice underwent intravital imaging. To visualize megakaryocytes and sinusoids, an AF488-conjugated anti-GPIX antibody derivative and Texas Red dextran 70 kDa were

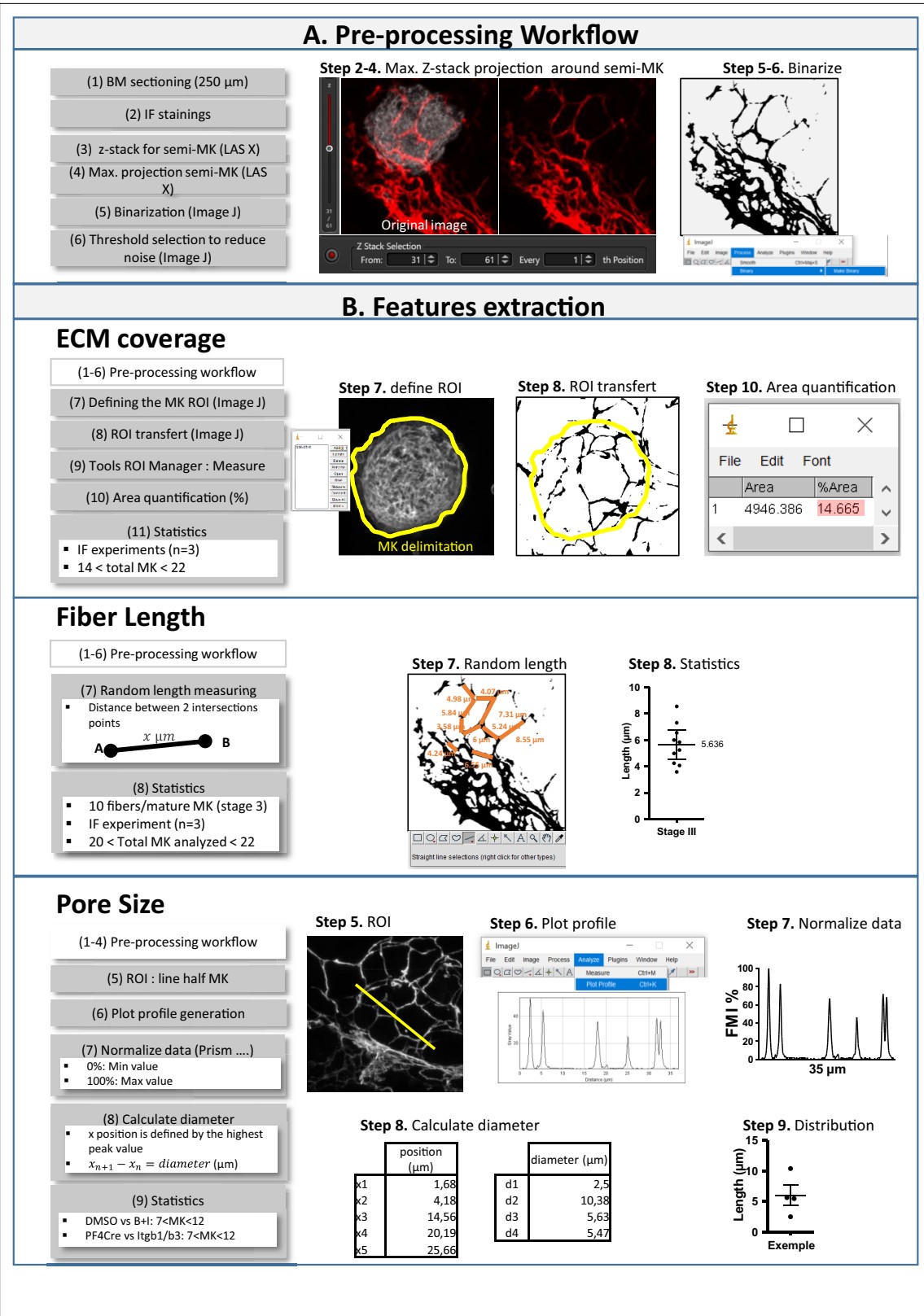

**Figure 8.** Image data flow graph.

intravenously injected, respectively. The skull bone marrow was observed using two-photon microscopy, following the procedure described in reference (*Bornert et al., 2021*). The anesthetized mice were monitored for a maximum of 6 hr, during which one to four proplatelets were recorded. Two regions of interest were analyzed, each composed of three images (xyz = 1320 µm*1320 µm*100 µm, total volume/ROI = 0.17 µm3, pixel size = 0.867 µm at obj HC FLUOTAR L 25 x/0.95 WATER zoom x1). A total of 130 megakaryocytes *Pf4cre* vs. 130 megakaryocytes *Itgb1$^{-/-}$/Itgb3$^{-/-}$* were analyzed and the results are expressed in %.

## Electron microscopy

For Transmission electron microscopy, bone marrow was fixed in 2.5% glutaraldehyde for 1 hr and embedded in Epon as described (*Scandola et al., 2021*). Transversal thin sections of the entire bone marrow were cut and examined under a JEOL TEM (120 kV). The number of megakaryocytes was counted per surface unit (s.u., 12,945 µm$^2$). The observations from three independent mice were averaged.

For SEM, native bone marrow megakaryocytes were allowed to adhere to a surface coated with 300 µg/ml fibronectin, 100 µg/ml fibrinogen, or 50 µg/ml laminin 511. After gentle agitation to detach non-adherent cells, the remaining adherent cells were fixed in 2.5% glutaraldehyde for 1 hr, dehydrated, attached to stubs, sputter coated, and examined under a Helios NanoLab microscope at 5kV (ThermoFisher, Eindhoven, The Netherlands). Adherent megakaryocytes were counted and classified as spreading or round cells. The results were obtained by the average from three independent mice.

## Flow cytometry

Flow cytometry was used to investigate (i) megakaryocyte ploidy, (ii) intact megakaryocytes in the blood, and (iii) platelet activation.

For the ploidy analysis of the native megakaryocytes, mouse femurs, tibias, and iliac crests were harvested, cut into small fragments, and incubated in a PBS-collagenase-dispase mix (at 3 mg/ml and 4 mg/ml, 5 ml/mouse, respectively) for 30 min at 37 °C. Then, the tube was filled with PBS-2% NCS to stop the collagenase, and the supernatant was 70 µm filtered to eliminate the bone fragments. After red cell lysis, the freshly isolated bone marrows were washed in PBS-2% NCS and stained with anti-CD41 and anti-CD42c antibodies and Hoechst. Anti Gr-1, B220, F4/80, and TER119 probes were used as a negative control to exclude granulocytes, B lymphocytes, macrophages, and erythrocytes from the total bone marrow suspension. The freshly isolated megakaryocytes were identified as a CD41/CD42c-positive cell population, and ploidy analysis was performed using Hoechst 33342. The results are representative of three independent experiments.

For detection of intact megakaryocytes in the blood, red blood cell lysis was performed using BD Lysing buffer solution according to manufacturer's instruction. The resulting mononucleated cells were labeled with fluorescent conjugated antibodies: anti CD41-PE-Cy7 (clone MWReg30), anti-CD42c-Alexa 488 (clone RAM-1), anti-CD45-PE (clone 30-F11), TER119-APC (clone TER119), antibodies from ebiosciences except RAM-1 antibody produced in house. Megakaryocytes enriched from bone marrow using BSA gradient were spiked into blood sample to validate the gating strategy to identify potential circulating megakaryocytes as CD45$^+$TER119$^-$CD41$^+$CD42c$^+$ events. 300.10$^3$ CD45$^+$TER119$^-$ events were recorded. Data acquisition was performed on a Symphony A1 flow cytometer (BD Biosciences) and analyzed using BD FACS Diva Software (BD Biosciences).

For platelet activation studies, whole blood was collected from the tail vein and anticoagulated with hirudin (200 U/ml). Platelets were activated or not with collagen-related peptide (CRP, 40 µg/ml) and TRAP (4 mM) for 15 min at 37 °C and stained with FITC-labeled or 647-labeled rat anti-mouse GPIbβ and APC-labeled rat anti-mouse P-selectin or PE-labeled rat anti-mouse activated-αIIbβ3 antibodies. Fluorescence was quantified using an LSRFortessa cell analyzer (BD Biosciences) and BD FACSDiva software. A total of 20,000 events were analyzed for each sample.

## Microfluidic experiments

The PDMS microfluidic chamber channels were assembled and connected to a peristaltic pump, as described (*Osmani et al., 2021*). The channels were washed with PBS for 5 min and coated with laminin 511 (50 µg/ml), fibrillar fibronectin (300 µg/ml), or fibrinogen (100 µg/ml) overnight at 4 °C, followed by a blocking stage with human serum albumin (1 %) for 30 min at RT. To produce

fibrillar fibronectin, we use the method of mechanical stretching as described in *Maurer et al., 2015*. Freshly isolated megakaryocytes were seeded at a concentration of 30 cells/µl per channel and incubated for 15 min – 45 min at 37 °C with 5% $CO_2$. In previous work, the seeding time was determined with capture efficiencies of over 80% for *Pf4cre* megakaryocytes (*Figure 4—figure supplement 1B*). The channels were then perfused under a flow of 300 µm/s and the megakaryo-cyte behavior was monitored in real-time for 1 min, using a Leica DMI8 microscope (x20) equipped with a CMOS Camera (Hamamatsu ORCA fusion). Megakaryocyte capture yield was measured by quantifying the number of adherent megakaryocytes before and after flow induction (expressed as a percentage) on an average of 20 stage positions (n=50 megakaryocytes analyzed/test). The results are an average of three independent experiments, and the values were expressed as a mean ± sem.

## Bone marrow explants

Preparation of bone marrow explants was performed as described in *Guinard et al., 2021* to investigate proplatelet formation. For investigating megakaryocyte emigration, bone marrows were flushed from femurs of *Pf4cre* and *Itgb1$^{-/-}$/Itgb3$^{-/-}$*; *Lama4$^{+/+}$* and *Lama4$^{-/-}$*; as well as from DMSO- and B+I-treated mice, and ten 0.5-mm-thick sections were placed in an incubation chamber. Megakaryocytes at the periphery of the explants were counted under an inverted phase contrast microscope coupled to a video camera (DMI8 Leica microscope, 40 x objective). A motorized multiposition stage (in *x, y, z*) was used, and an average of 100 stages positions showing megakaryocytes was followed. Results are expressed as the number of total emigrating megakaryocytes at the indicated time points. In each case, a minimum of six independent experiments were performed. Mice from each genotype were analyzed and image acquisitions were performed in a blinded manner.

## Effects of MMP inhibition

Batimastat plus ilomastat treatment was tested for their potential impact on the ECM in the micro-environment of the megakaryocyte vascular niche. Mice were treated daily with a protease inhibitor cocktail (Batimastat at 30 mg/kg+Ilomastat at 50 mg/kg) or vehicle (DMSO) for 7 consecutive days (*Gui et al., 2018*; *Pielecka-Fortuna et al., 2015*). The platelet count was monitored every 2 days. On the eighth day, the mice were sacrificed, and their bone marrow was collected for analysis of ECM organization and megakaryocyte behavior, as explained in the 3D confocal analysis on bone marrow vibratome sections.

## Gelatin degradation assay

Coverslips were coated with Oregon green gelatin and fixed with 0.5% glutaraldehyde for 20 min at RT. After washing three times with PBS, cells were seeded on coated coverslips and incubated for 6 hr before fixation and staining.

## Statistics

All values are reported as the mean ± sem. n=number of megakaryocytes studied. Statistical analyses were performed with PrismGraphpad software (La Jolla, CA, USA). For group comparison, data were tested for Gaussian distribution. Then, a Student t-test (Gaussian) or Mann-Whitney U test (non-Gaussian) was used to compare individual groups; multiple groups were compared by one-way ANOVA followed by Bonferroni post-test or by a non-parametric Kruskal-Wallis test, with a threshold of significance of 5%. p-Values <0.05 were considered statistically significant. *p<0.05; **p<0.01; ***p<0.001.

## Acknowledgements

We wish to thank Ketty Knez-Hippert, Josiane Weber, Clarisse Mouriaux, Sylvie Moog and Patricia Laeuffer (EFS-GEST) for excellent expert technical assistance. This work was supported by ARMESA (Association de Recherche et Développement en Médecine et Santé Publique) and by ANR Grant MegaPod (ANR-22-CE14-0029). The authors thank Pr. B Nieswandt for providing *Gp6* knock-out mice.

## Additional information

### Funding

| Funder | Grant reference number | Author |
|---|---|---|
| Agence Nationale de la Recherche | ANR-22-CE14-0029 | Claire Masson<br>Renaud Poincloux<br>Olivier Destaing<br>Anita Eckly |
| Association de Recherche et Développement en Médecine et Santé Publique | | Anita Eckly |

The funders had no role in study design, data collection and interpretation, or the decision to submit the work for publication.

### Author contributions

Claire Masson, Investigation, Visualization, Methodology, Writing – original draft, Project administration, Writing – review and editing; Cyril Scandola, Jean-Yves Rinckel, Investigation, Visualization, Methodology; Fabienne Proamer, Emily Janus-Bell, Alma Mansson, Investigation, Methodology; Fareeha Batool, Formal analysis; Naël Osmani, Léa Mallo, Nathalie Brouard, Catherine Leon, Alicia Bornert, Methodology; Jacky G Goetz, Renaud Poincloux, Olivier Destaing, Writing – original draft; Hong Qian, Writing – original draft, Writing – review and editing; Maxime Lehmann, Funding acquisition, Writing – original draft, Writing – review and editing; Anita Eckly, Conceptualization, Funding acquisition, Investigation, Visualization, Methodology, Writing – original draft, Project administration, Writing – review and editing

### Author ORCIDs
Claire Masson ⓘ https://orcid.org/0000-0003-4249-0450
Cyril Scandola ⓘ https://orcid.org/0000-0002-5305-9095
Naël Osmani ⓘ https://orcid.org/0000-0003-1195-753X
Jacky G Goetz ⓘ https://orcid.org/0000-0003-2842-8116
Renaud Poincloux ⓘ https://orcid.org/0000-0003-2884-1744
Alma Mansson ⓘ https://orcid.org/0009-0006-1982-7712
Anita Eckly ⓘ https://orcid.org/0000-0001-9620-4961

### Ethics
All animal studies were approved by the French legislation for animal experimentation and in accordance with the guide for the care and use of laboratory animals as defined by the European laws (Animal Facility Agreement C-67-482-10).

Reviewer #1 (Public review): https://doi.org/10.7554/eLife.104963.4.sa1
Reviewer #2 (Public review): https://doi.org/10.7554/eLife.104963.4.sa2
Author response https://doi.org/10.7554/eLife.104963.4.sa3

## Additional files

### Supplementary files
MDAR checklist

### Data availability
Figure source data are provided.

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
