## [Editor Report · eLife Assessment]

In this revised version, the authors provide a thorough investigation of the interaction of megakaryocytes (MK) with their associated extracellular matrix (ECM) during maturation; they provide **compelling** evidence that the existence of a dense cage-like pericellular structure containing laminin γ1 and α4 and collagen IV is key to fixing the perisinusoidal localization of MK and preventing their premature intravasation. Adhesion of MK to this ECM cage is dependent on integrin beta1 and beta3 expressed by MK. This strong conclusion is based on the use of state-of-the art techniques such f primary murine bone marrow MK cultures, mice lacking ECM receptors, namely integrin beta1 and beta3 null mice, as well as high-resolution 2D and 3D imaging. The study provides **valuable** insight into the role of cell-matrix interactions in MK maturation and provides an interesting model with practical implications for the fields of hemostasis and thrombosis.

---

## [Referee Report · Reviewer #1 (Public review)]

The authors report on a thorough investigation of the interaction of megakaryocytes (MK) with their associated ECM during maturation. They report convincing evidence to support the existence of a dense cage-like pericellular structure containing laminin γ1 and α4 and collagen IV, which interacts with integrins β1 and β3 on MK and serve to fix the perisinusoidal localization of MK and prevent their premature intravasation. As with everything in nature, the authors support a Goldilocks range of MK-ECM interactions - inability to digest the ECM via inhibition of MMPs leads to insufficient MK maturation and development of smaller MK. This important work sheds light into the role of cell-matrix interactions in MK maturation, and suggests that higher-dimensional analyses are necessary to capture the full scope of cellular biology in the context of their microenvironment. The authors have responded appropriately to the majority of my previous comments.

---

## [Referee Report · Reviewer #2 (Public review)]

Summary:

This study makes a significant contribution to understanding the microenvironment of megakaryocytes (MKs) in the bone marrow, identifying an extracellular matrix (ECM) cage structure that influences MK localization and maturation. The authors provide compelling evidence for the presence of this ECM cage and its role in MK homeostasis, employing an array of sophisticated imaging techniques and molecular analyses.

The authors have addressed most of the concerns raised in the previous review, providing clarifications and additional data that strengthen their conclusions

More broadly, this work adds to a growing recognition of the ECM as an active participant in haematopoietic cell regulation in the bone marrow microenvironment. This work could pave the way to future studies investigating how the megakaryocytes' ECM cage affects their function as part of the haematopoietic stem cell niche, and by extension, influences global haematopoiesis.

---

## [Author Response]

The following is the authors’ response to the previous reviews.

**Minor Issues:**
(1) As the authors mention, MKs have been suggested to mature rapidly at the sinusoids, and both integrin KO and laminin KO MKs appear mislocalized away from the sinusoids. Additionally, average MK distances from the sinusoid may also help separate whether the maturation defects could be in part due to impaired migration towards CXCL12 at the sinusoid. Presumably, MKs could appear mislocalized away from the sinusoid given the data presented suggesting they are leaving the BM and entering circulation. Additional commentary on intrinsic (ex-vivo) MK maturation phenotypes may help strengthen the author's conclusions

Thank you for your insightful suggestion regarding intrinsic MK maturation defects in integrin KO and laminin KO mice. This indeed could be the case. We have now addressed this possibility in the revised discussion section (page 14; lines 14-15), acknowledging intrinsic maturation defects as a potential contributor to observed maturation issues.

(2) It would be helpful if the authors could comment as to whether MKs are detectable in blood.

We appreciate the opportunity to clarify this point. Intact Itgb1^-/-^/Itgb3^-/-^ MKs were not detected in the peripheral blood by either flow cytometry or blood smear analysis. This indicates that megakaryocytes do not normally circulate in the systemic bloodstream. Instead, we observed large MK nuclei trapped specifically within the lung capillaries, consistent with their known physical retention in the pulmonary circulation during platelet release. This explanation is now better explained on page 10, lines 14-19.

(3) Supplementary Figure 6 - shows no effect on in vitro MK maturation and proplt, or MK area - But Figures 6B/6C demonstrate an increase in total MK number in MMP-inhibitor treated mice compared to control. This discrepancy should be better discussed.

We have now expanded the discussion in the revised manuscript to address the different results obtained in vitro and in vivo, emphazing that the in vitro model may not fully recapitulate the complex and dynamic bone marrow ECM niche. Additionally, differences in the source and regulation of MMPs likely contribute to the differing outcomes, underlining the importance of studying these processes within their physiological context. For instance, non-megakaryocytic sources of MMPs and paracrine regulatory mechanisms may play a critical role within the physiological microenvironment, ultimately affecting MK proliferation and maturation in a manner not observed in simplified culture systems. This clarifications can be found on page 12, lines 6-17.

(4) A function of the ECM discussed relates to MK maturation but in the B1/3 integrin KO mice, the presence of the ECM cage is reduced but there appears to be no significant impact upon maturation (Supplementary Figure 4). By contrast, MMP inhibition in vivo (but not in vitro) reduces MK maturation. These data could be better clarified in the text.

Thank you for raising this important point. While Suppl. Figure 4 shows normal size and ploidy in DKO MK, a critical defect is revealed at the ultrastructural level. Mature DKO MKs exhibit severe dysplasia of the demarcation membrane system (DMS), characterized by extensive membrane accumulation and abnormal archirecture, with no typical platelet territories visible. This DMS defect directly impairs MK maturation and explains the thrombocytopenia observed in these mice. Increased emperipolesis further indicated disrupted maturation processes. These observations confirm the essential role of the ECM cage in supporting proper DMS organization and overall MK maturation in vivo, consistent with findings from MMP inhibition experiments. We have clarified and emphasized the significance of these DMS abnormalities in the revised manuscripts, including updated results (Page 9, lines 17-21) and a new EM image in Suppl. Figure 4.

**Reviewer #1 (Public review):**
The authors report on a thorough investigation of the interaction of megakaryocytes (MK) with their associated ECM during maturation. They report convincing evidence to support the existence of a dense cage-like pericellular structure containing laminin γ1 and α4 and collagen IV, which interacts with integrins β1 and β3 on MK and serve to fix the perisinusoidal localization of MK and prevent their premature intravasation. As with everything in nature, the authors support a Goldilocks range of MK-ECM interactions - inability to digest the ECM via inhibition of MMPs leads to insufficient MK maturation and development of smaller MK. This important work sheds light into the role of cell-matrix interactions in MK maturation, and suggests that higher-dimensional analyses are necessary to capture the full scope of cellular biology in the context of their microenvironment. The authors have responded appropriately to the majority of my previous comments.

We sincerely thank the reviewer for their insightful comments.

Some remaining points:In a previous critique, I had suggested that "it is unclear how activation of integrins allows the MK to become "architects for their ECM microenvironment" as the authors posit. A transcriptomic analysis of control and DKO MKs may help elucidate these effects". The authors pointed out the technical difficulty of obtained sufficient numbers of MK for such analysis, which I accept, and instead analyzed mature platelets, finding no difference between control and DKO platelets. This is not necessarily surprising, since mature circulating platelets have no need to engage an ECM microenvironment, and for the same reason I would suggest that mature platelet analyses are not representative of MK behavior as regards ECM interactions.

We fully agree with the reviewer that platelet analyses do not accurately reflect the behavior of MKs in the context of interactions with the ECM. This understanding is also one of the reasons why we chose not to include RT-PCR data on platelets in our manuscript. Instead, we emphasize the role of integrins as essential regulators of ECM remodeling, as they transmit traction forces that can significantly influence this process. We also report reduced RhoA activation in DKO MK, which is likely to affect ECM organization. We believe that these explanations contribute to a clearer understanding of how integrin activation enables megakaryocytes to act as "architects" of their ECM microenvironment.

**Reviewer #2 (Public review):**
This study makes a significant contribution to understanding the microenvironment of megakaryocytes (MKs) in the bone marrow, identifying an extracellular matrix (ECM) cage structure that influences MK localization and maturation. The authors provide compelling evidence for the presence of this ECM cage and its role in MK homeostasis, employing an array of sophisticated imaging techniques and molecular analyses.The authors have addressed most of the concerns raised in the previous review, providing clarifications and additional data that strengthen their conclusion.More broadly, this work adds to a growing recognition of the ECM as an active participant in haematopoietic cell regulation in the bone marrow microenvironment. This work could pave the way to future studies investigating how the megakaryocytes' ECM cage affects their function as part of the haematopoietic stem cell niche, and by extension, influences global haematopoiesis.

We thank this reviewer for providing such constructive feedback.